# Efficient and Effective Augmentation Strategy for Adversarial Training

**Sravanti Addepalli**[†][*]     **Samyak Jain**[†][◇][‡][*]     **R.Venkatesh Babu**[†]
[†] Video Analytics Lab, Indian Institute of Science, Bangalore
[◇] Indian Institute of Technology (BHU) Varanasi

## Abstract

Adversarial training of Deep Neural Networks is known to be significantly more data-hungry when compared to standard training. Furthermore, complex data augmentations such as AutoAugment, which have led to substantial gains in standard training of image classifiers, have not been successful with Adversarial Training. We first explain this contrasting behavior by viewing augmentation during training as a problem of domain generalization, and further propose Diverse Augmentation-based Joint Adversarial Training (DAJAT) to use data augmentations effectively in adversarial training. We aim to handle the conflicting goals of enhancing the diversity of the training dataset and training with data that is close to the test distribution by using a combination of simple and complex augmentations with separate batch normalization layers during training. We further utilize the popular Jensen-Shannon divergence loss to encourage the *joint* learning of the *diverse augmentations*, thereby allowing simple augmentations to guide the learning of complex ones. Lastly, to improve the computational efficiency of the proposed method, we propose and utilize a two-step defense, Ascending Constraint Adversarial Training (ACAT), that uses an increasing epsilon schedule and weight-space smoothing to prevent gradient masking. The proposed method DAJAT achieves substantially better robustness-accuracy trade-off when compared to existing methods on the RobustBench Leaderboard on ResNet-18 and WideResNet-34-10. The code for implementing DAJAT is available here: https://github.com/val-iisc/DAJAT.

## 1   Introduction

Deep Neural Network (DNN) based image classifiers are vulnerable to crafted imperceptible perturbations known as Adversarial Attacks [41] that can flip the predictions of the model to unrelated classes leading to disastrous implications. Adversarial Training [14, 27, 50] has been the most successful defense strategy, where a model is explicitly trained to be robust in the presence of such attacks. While early defenses focused on designing suitable loss functions for training, subsequent works [29, 34] showed that with careful hyperparameter tuning, even the two most popular methods PGD-AT [27] and TRADES [50] yield comparable performance, highlighting the saturation in performance with respect to changes in the training loss. Schmidt et al. [35] observed that adversarial training has a large sample complexity and further gains require the use of additional training data. Subsequent works [5, 16] indeed used additional data whose distribution is close to that of the original dataset in order to obtain performance gains. The availability of large amounts of relevant data is impractical to assume, leading to an exploration towards augmentations based on Generative Adversarial Networks [13] and Diffusion based models [19, 16]. However, the use of such generative models incurs an

---

[*]Equal Contribution.
Correspondence to Sravanti Addepalli <sravantia@iisc.ac.in>, Samyak Jain <samyakjain.cse18@itbhu.ac.in>
[‡] Work done during internship at Video Analytics Lab, Indian Institute of Science.

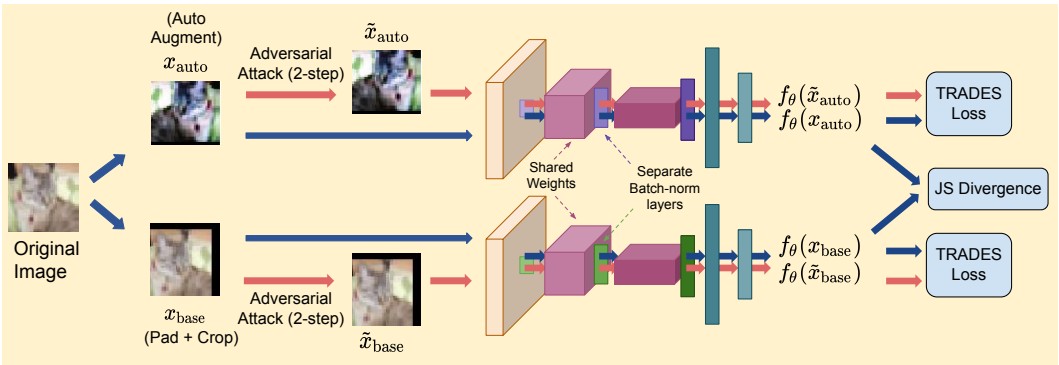

Figure 1: A Schematic representation of the proposed approach DAJAT

additional training cost and suffers from limited diversity in low-data regimes and in datasets with high-resolution images.

A simple and efficient solution to improve the diversity of training data in standard Empirical Risk Minimization (ERM) based training has been the use of random transformations such as rotation, color jitter, and variations in contrast, sharpness and brightness [24, 8, 9], which can change images significantly in input space while belonging to the same class as the original image. However, prior works have surprisingly found that such augmentations, that cause large changes in the input distribution, do not help adversarial training [34, 15, 40]. This limits the augmentations in adversarial training to simple ones - zero padding followed by random crop, and horizontal flip [34, 29, 15] - which may not be able to fill in the large data requirement of Adversarial Training.

In this work, we firstly analyze the reasons for this contrasting trend between Standard and Adversarial Training, and further show that **it is indeed possible to utilize complex augmentations effectively in Adversarial training as well**, by jointly training on simple and complex data augmentations using separate batch-normalization layers for each type, as shown in Fig.1. While complex augmentations increase the data diversity resulting in better generalization, simple augmentations ensure that the model specializes on the training data distribution as well. We further minimize the Jenson-Shannon divergence between the softmax outputs of various augmentations to enable the simple augmentations to guide the learning of complex ones. In order to improve the computational efficiency of the proposed method, we use two attack steps (instead of 10) during training. By progressively increasing the magnitude of perturbations and performing smoothing in weight space, we show that it is indeed possible to improve the stability of training. Our contributions are listed below:

- We analyze the reasons for the failure of strong data augmentations in adversarial training by viewing augmentation during training as a domain generalization problem, and further propose *Diverse Augmentation based Joint Adversarial Training* (DAJAT) to utilize data augmentations effectively in Adversarial training. The proposed approach can be integrated with many augmentations and adversarial training methods to obtain performance gains.

- We propose and integrate DAJAT with an efficient 2-step defense strategy, *Ascending Constraint Adversarial Training* (ACAT) that uses linearly increasing $\varepsilon$ schedule, cosine learning rate and weight-space smoothing to prevent gradient masking and improve convergence.

- We obtain improved robustness and large gains in standard accuracy on multiple datasets (CIFAR-10, CIFAR-100, ImageNette) and model architectures (RN-18, WRN-34-10). We obtain remarkable gains in a low data scenario (CIFAR-100, Imagenette) where data augmentations are most effective. **On CIFAR-100, we outperform all existing methods on the RobustBench leaderboard [7] with the same model architecture.**

## 2 Related Works

We discuss various existing strategies for improving the Adversarial robustness of Deep Networks.

**Adversarial Training (AT):** Goodfellow et al. [14] proposed FGSM-AT where single-step adversarial samples were used for training. However, these models were susceptible to gradient masking [30], where the local loss landscape becomes convoluted leading to the generation of weaker single-step

attacks during training. This leads to a false sense of security against single-step attacks, while the models are still susceptible to stronger multi-step attacks such as PGD [27]. PGD-AT [27, 34] used multi-step attacks in a similar adversarial training formulation to obtain robust models that stood the test of time against several attacks [2, 6, 38]. TRADES [50] explicitly optimizes the trade-off between the accuracy on natural and adversarial examples by minimizing the cross-entropy loss on natural images along with the Kullback-Leibler (KL) divergence between the predictions of adversarial and clean images. In the proposed defense, we use the base loss from TRADES-AT [50].

Several works have explored the use of auxiliary techniques in AT such as weight-space smoothing [22, 44], architectural changes [15, 47] and increasing the diversity of training data by using additional natural and synthetic data [35, 5, 15, 33]. Increasing the diversity of training data achieves significant performance gains since the sample complexity of adversarial training is known to be high [35].

**Augmentations in Adversarial Training:** While data augmentations such as contrast, sharpness and brightness adjustments are known to improve performance in the standard training regime, they have not led to substantial gains in adversarial training. AutoAugment [8] uses Proximal Policy Optimization to find the set of policies that can yield optimal performance on a given dataset. Contrary to prior works [34, 15], we show that the policies that are optimized for standard training indeed yield a boost in performance of adversarial training as well, when used in the proposed training framework.

In a recent work, Rebuffi et al. [33] show that it is possible to obtain substantial gains in robust accuracy by using spatial composition based augmentations such as CutMix [48] and CutOut [11] that preserve low-level features of the image. Cutmix replaces part of an image with another and also combines the output softmax vectors in the same ratio, while CutOut blanks out a random area of an image. The authors hypothesize that the augmentations used in Adversarial training need to preserve low-level features, which severely limits the possibilities for mitigating the large sample complexity in adversarial training. We show that by using the proposed approach DAJAT, it is indeed possible to use augmentations such as color jitter, contrast, sharpness and brightness adjustments that significantly change the low level statistics of images (Ref: Table-6).

# 3   Preliminaries: Notation and Threat Model

We consider the Adversarial Robustness of DNN based image classifiers. An input image is denoted as $x \in \mathcal{X}$ and the corresponding ground truth label as $y \in [0, 1]$. We denote a simple transformation of $x$ obtained using Pad, Crop and Horizontal flip (Pad+Crop+HFlip, referred to as Base augmentations) by $x_{\text{base}}$ and other transformations of $x$ by the respective subscript. For example, $x_{\text{auto}}$ refers to the image $x$ being transformed by AutoAugment (AA) [8] followed by the base augmentations. The function mapping of the classifier $C$ from input space $\mathcal{X}$ to the softmax vectors is denoted using $f_\theta(.)$, where $\theta$ denotes the network parameters. Adversarial examples corresponding to the images $x$, $x_{\text{base}}$ and $x_{\text{auto}}$ are denoted using $\tilde{x}$, $\tilde{x}_{\text{base}}$ and $\tilde{x}_{\text{auto}}$ respectively. We consider the $\ell_\infty$ norm based threat model, where $\tilde{x}$ is a valid perturbation within $\varepsilon$ if it belongs to the set $\mathcal{A}_\varepsilon(x) = \{\tilde{x} : ||\tilde{x} - x||_\infty \le \varepsilon\}$.

# 4   Motivation: Role of Augmentations in Neural Network Training

In this section, we first explore the contrasting factors that influence the training of neural networks when data is augmented (Sec.4.1), and further delve into the specifics of adversarial training which make it challenging to obtain gains using complex data augmentations (Sec.4.2).

## 4.1   Impact of Augmentations in Neural Network Training

**Conjecture-1:** We hypothesize that the role of data augmentations in the training of Neural Networks is influenced by the following contrasting factors:

  (i) Reduced overfitting due to an increase in diversity of the augmented dataset, leading to better generalization of the network to the test set.

 (ii) Larger domain shift between the augmented data distribution and the test data distribution, leading to a drop in performance on the test set.

(iii) Capacity of the Neural Network in being able to generalize well to the augmented data distribution and the unaugmented data distribution for the given task.

**Justification:** The training of Neural Networks using augmented data can be considered as a problem of domain generalization, where the network is trained on a source domain (augmented data) and is

Table 1: **Impact of augmentations:** Performance (%) of ACAT models on Base augmentations and AutoAugment (Auto). Clean and robust accuracy against GAMA attack [38] are reported. The use of AutoAugment results in $\sim 1.5 - 2\%$ drop in robust accuracy.

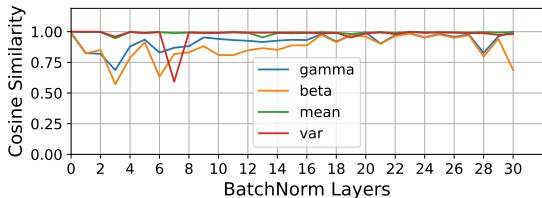

| Model | Test:
Train ↓ | No Aug
Clean | 
Robust | AutoAugment
Clean | 
Robust |
|---|---|---|---|---|---|
| ResNet-18 | Base | 82.41 | **50.00** | 63.79 | 37.07 |
| | Auto | **82.54** | 48.11 | **76.40** | **43.22** |
| WideResNet-34-10 | Base | 86.71 | **55.58** | 68.24 | 40.83 |
| | Auto | **86.80** | 53.99 | **82.64** | **48.98** |

Figure 2: **Comparison of BN layer statistics** for a WRN-34-10 model trained on CIFAR-10 using DAJAT. BN layers of the Base augmentations (Pad+Crop,H-Flip) are compared with those of AutoAugment. Initial layer (L3) parameters are diverse, while those of deeper layers (L25) are similar.

expected to generalize to a target domain (test data). We use the theoretical formulation by Ben-David et al. [3] shown below to justify the respective claims in Conjecture-1:

$$\epsilon_t(f) \leq \epsilon_s(f) + \frac{1}{2}d_{\mathcal{F}\Delta\mathcal{F}}(s,t) + \lambda \tag{1}$$

(i) The use of a more diverse or larger source dataset reduces overfitting, improving the performance of the network on the source distribution. From Eq.1, expected error on the target distribution $\epsilon_t$ (test set in this case) is upper bounded by the expected error on the source distribution $\epsilon_s$ (augmented dataset) along with other terms. Therefore, improved performance on the augmented distribution can improve the performance on the test set as well.

(ii) The expected error on the target distribution $\epsilon_t$ is upper bounded by the distribution shift between the source and target distributions $\frac{1}{2}d_{\mathcal{F}\Delta\mathcal{F}}(s,t)$ along with other terms. Therefore, a larger domain shift between the augmented and test data distributions can indeed limit the performance gains on the test set.

(iii) The constant $\lambda$ in Eq.1 measures the risk of the optimal joint classifier: $\lambda = \min_{f \in \mathcal{F}} \epsilon_s(f) + \epsilon_t(f)$.

Neural Networks with a higher capacity can minimize the expected risk on the source set $\epsilon_s$ and the risk of the optimal joint classifier effectively. Therefore, capacity of the Neural Network and complexity of the task influence the gains that can be obtained using augmentations.

### 4.2 Analysing the role of Augmentations in Adversarial Training

We analyse the trade-off between the factors described in Conjecture-1 for adversarial training when compared to standard ERM training. In addition to the goal of improving accuracy on clean samples, adversarial training aims to achieve local smoothness of the loss landscape as well. Hence, the complexity of Adversarial Training is higher than that of standard ERM training, making it important to use larger model capacities to obtain gains using data augmentations (based on Conjecture-1 (iii)). This justifies the gains obtained by Rebuffi et al. [33] on the WRN-70-16 architecture by using CutMix based augmentations (2.9% higher robust accuracy and 1.23% higher clean accuracy). The same method does not obtain significant gains on smaller architectures such as ResNet-18 where a 1.76% boost in robust accuracy is accompanied by a 2.55% drop in clean accuracy.

Secondly, while the distribution shift between augmented data and test data ($\frac{1}{2}d_{\mathcal{F}\Delta\mathcal{F}}(s,t)$) may be sufficiently low for natural images leading to improved generalization to test set (Conjecture-1(i,ii)), the same may not be true for adversarial images. There is a large difference between the augmented data and test data in pixel space, although they may be similar in feature space. Since adversarial attacks perturb images in pixel space, the distribution shift between the corresponding perturbations widens further as shown in Fig.3-5 of the Appendix (Details in Appendix-B). Based on Conjecture-1(ii), unless this difference is accounted for, complex augmentations cannot improve the performance of adversarial training. This trend has also been observed empirically by Rebuffi et al. [33], where they conclude that augmentations designed for robustness need to preserve low-level features.

We present the performance of Adversarial Training by using either Base augmentations (Pad+Crop, Flip) or AutoAugment [8] during training and inference on the CIFAR-10 dataset using ResNet-18 and WideResNet-34-10 architectures in Table-1. Firstly, we note that by using AutoAugment during training alone, robust accuracy on the test set drops by $\sim 1.5 - 2\%$ which is as observed in prior work [15]. Secondly, the clean and robust accuracy drop by $\sim 6.5\%$ when augmented images are

used for both training and testing, highlighting the complexity of the learning task. We present additional results by training without any augmentation, and by training using a combination of both augmentations in every minibatch in Table-9 of the Appendix. The use of Base Augmentations alone (Pad+Crop+HFlip) still gives the best overall performance on the unaugmented test set.

## 5 Proposed Method

### 5.1 Background

We briefly discuss the TRADES-AWP defense [50, 44], which is the base algorithm used in DAJAT.

$$\mathcal{L}_{\text{AWP}} = \max_{\hat{\theta} \in \mathcal{M}(\theta)} \frac{1}{N} \sum_{i=1}^{N} \mathcal{L}_{\text{CE}}(f_{\theta+\hat{\theta}}(x_i), y_i) + \beta \cdot \max_{\tilde{x}_i \in \mathcal{A}_\varepsilon(x_i)} \text{KL}(f_{\theta+\hat{\theta}}(x_i) || f_{\theta+\hat{\theta}}(\tilde{x}_i)) \quad (2)$$

Firstly, an adversarial attack is generated by maximizing the KL divergence between the softmax predictions of the clean and adversarial examples iteratively for 10 attack steps. An Adversarial Weight Perturbation (AWP) step additionally perturbs the weights of the model to maximize the overall loss in weight-space. The weight perturbations are constrained in the feasible region $\mathcal{M}(\theta)$ such that for a given layer $l$, $||\hat{\theta}_l|| \leq \gamma \cdot ||\theta_l||$. The overall training loss is a combination of Cross-entropy loss on clean samples and the KL divergence term. The latter is weighted by a factor $\beta$ that controls the robustness-accuracy trade-off. Training the model using Adversarial Weight Perturbations leads to smoothing of loss surface in the weight space, resulting in better generalization [44, 40].

### 5.2 Diverse Augmentation based Joint Adversarial Training (DAJAT)

As discussed in Section-4, the use of augmentations in training can be viewed as a problem of domain generalization, where performance on the source distribution or augmented dataset is crucial towards improving the performance on the target distribution or test set. Since adversarial training is inherently challenging, for limited model capacity, it is difficult to obtain good performance on the training data that is transformed using complex augmentations. Moreover, the large distribution shift between augmented data and test data, specifically with respect to low-level statistics, results in poor generalization of robust accuracy to the test set.

To mitigate these challenges, we propose the combined use of simple and complex augmentations during training, so that the model can benefit from the diversity introduced by complex augmentations, while also specializing on the original data distribution that is similar to the simple augmentations. We propose to use separate batch normalization layers for simple and complex augmentations, so as to offset the shift in distribution between the two kinds of augmentations. In our main approach, we propose to use Pad and Crop followed by Horizontal Flip (Pad+Crop+HFlip) as the Simple augmentations, and Autoaugment followed by Pad+Crop+HFlip as the complex augmentations. We justify the choice of this augmentation pipeline in Appendix-B.3.

Motivated by AugMix [18], we additionally minimize the Jenson-Shannon (JS) divergence between the softmax outputs of different augmentations, so as to allow the simple augmentations to guide the learning of complex ones. We present the training loss $\mathcal{L}_{\text{DAJAT}}$ of the proposed method, Diverse Augmentation based Joint Adversarial Training (DAJAT) below in Eq.5:

$$\mathcal{L}_{\text{TR}}(\theta, x, y) = \mathcal{L}_{\text{CE}}(f_\theta(x), y) + \beta \cdot \max_{\tilde{x} \in \mathcal{A}_\varepsilon(x)} \text{KL}(f_\theta(x) || f_\theta(\tilde{x})) \quad (3)$$

$$\tilde{\theta} = \underset{\hat{\theta} \in \mathcal{M}(\theta)}{\text{argmax}} \frac{1}{N} \sum_{i=1}^{N} \mathcal{L}_{\text{TR}}(\theta + \hat{\theta}, x_{i,\text{base}}, y_i) \quad (4)$$

$$\mathcal{L}_{\text{DAJAT}} = \frac{1}{T+1} \cdot \frac{1}{N} \sum_{i=1}^{N} \left\{ \mathcal{L}_{\text{TR}}(\theta + \tilde{\theta}, x_{i,\text{base}}, y_i) + \sum_{t=1}^{T} \mathcal{L}_{\text{TR}}(\theta + \tilde{\theta}, x_{i,\text{auto(t)}}, y_i) \right\} +$$
$$\frac{1}{N} \sum_{i=1}^{N} \text{JSD}(f_{\theta+\tilde{\theta}}(x_{i,\text{base}}), f_{\theta+\tilde{\theta}}(x_{i,\text{auto(1)}}), \ldots, f_{\theta+\tilde{\theta}}(x_{i,\text{auto(T)}})) \quad (5)$$

Adversarial attacks are generated individually for each augmentation by maximizing the respective KL divergence term of the TRADES loss shown in Eq.3. To improve training efficiency, we compute $\tilde{x}$ using two attack steps with a step-size of $\varepsilon$. We use a combination of a linearly increasing schedule

of $\varepsilon$, cosine learning rate schedule and model weight-averaging [22] to improve the stability and performance of adversarial training (Details in Sec.5.4). The DAJAT loss (Eq.5) is a combination of the TRADES 2-step loss on each of the augmentations $x_{\mathrm{base}}$ and $x_{\mathrm{auto(t)}}$, along with an adversarial weight perturbation step on the loss corresponding to the base augmentations alone to reduce compute. For every batch normalization layer, two sets of running statistics and affine parameters are maintained and used for simple and complex augmentations respectively (Ref: Algorithm-1 in the Appendix).

While we use AutoAugment in our main approach, we show in Table-6 that the proposed approach works well with other augmentations as well. The role of the base (primary) augmentations is primarily to learn the batch normalization layers that would be used during inference time, and to provide better supervision for the training of complex augmentations using the JS divergence term. The role of the complex (secondary) augmentations is to enhance the diversity of the training dataset. Therefore we use a single primary augmentation and multiple instances of the secondary augmentation. We find that the gains in performance saturate with the addition of more instances of secondary augmentation, and therefore recommend the use of a single base augmentation and two instances of secondary augmentation for the best performance-accuracy trade-off. We note from Table-2 and Appendix-F.6 that in this setting, the computational efficiency of the proposed method is better than the TRADES-AWP [50, 44] defense, while achieving considerable performance gains.

## 5.3  Split Batch Normalization Layers for Different Augmentations

A Batch Normalization (BN) layer [21] is implemented as follows on a given feature map $g(x_i)$ of an input image $x_i$: $\hat{g}(x_i) = \frac{g(x_i)-\mu}{\sigma} \cdot \gamma + \beta$. Here, $\mu$ and $\sigma$ denote the mean and standard deviation of the current mini-batch during training. During inference, these are set to the running mean and variance computed during training. $\gamma$ and $\beta$ constitute parameters of the network that are trained.

Prior works use separate batch normalization layers to improve the performance of standard [46, 43, 28] and adversarial training [45, 42], in both supervised [46, 43, 28, 45, 42] and self-supervised settings [23, 12]. The proposed defense DAJAT uses separate batch normalization layers for simple and complex augmentations as discussed in the previous section. We do not use separate batch normalization for clean and adversarial images here.

In DAJAT, we maintain two sets of BN statistics $\mu$ and $\sigma$, and two sets of affine parameters, $\beta$ and $\gamma$ for every BN layer. We plot the cosine similarity between the BN layers corresponding to Base augmentations and AutoAugment across different layers on a WideResNet-34-10 model trained using DAJAT on CIFAR-10 in Fig.2. While the mean and variance are relatively more similar across most of the layers (0-30 in x-axis), we note significant differences in the $\gamma$ and $\beta$ values, specifically in the initial layers. We present a detailed study on the importance of having separate running statistics and affine parameters in Appendix-C, and conclude that our method works effectively by separating either the running statistics or affine parameters individually as well. However, it works best by using a combination of both (Ref: Table-5 of the Appendix). We further note that very small differences in running statistics (as seen in Fig.2 and Fig.7 of the Appendix) can also result in noteworthy performance gains. Finally, we find that the use of a single batch-norm layer for both augmentations degrades results significantly. Therefore, the use of different parameters for each augmentation type ensures that the function mapping differs for each augmentation, thereby effectively offsetting the large differences in low level statistics.

## 5.4  Ascending Constraint Adversarial Training (ACAT)

In this section, we discuss the methods incorporated to improve the training efficiency of DAJAT in greater detail. We apply these methods to the TRADES-AWP defense to independently analyse their impact, and term the proposed defense as Ascending Constraint Adversarial Training (ACAT). We aim to improve the training efficiency by reducing the number of attack steps of the base defense from 10 to 2. We use two attack steps for training since it is known to be more stable when compared to single-step adversarial training, while still being computationally efficient [39].

As shown in Fig.3, naively reducing the number of attack steps to 2 in TRADES-AWP AT (Fixed constraint AT) causes a drop in clean and robust accuracy. While the drop is larger at higher training $\varepsilon$, a drop in clean accuracy is seen at $\varepsilon = 8/255$ as well. Further, the large robustness gap between last and best epochs indicates that the training stability deteriorates towards the end of training. The instability of TRADES-AWP AT at $\varepsilon = 8/255$ is higher for larger model capacities (WideResNet-34-10) as discussed in Appendix-E.1.

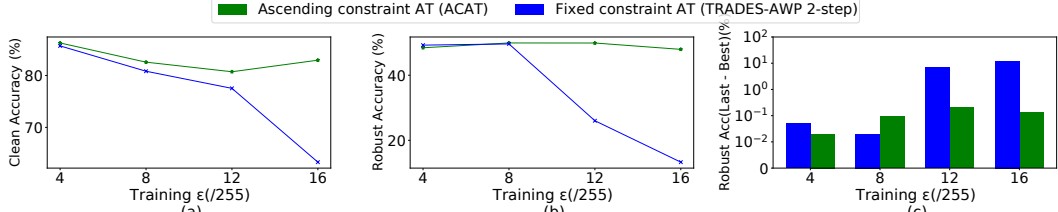

Figure 3: **Comparison of ACAT (2-step) against TRADES-AWP 2-step baseline** on CIFAR-10 with ResNet-18 architecture. (a) Clean accuracy, (b) Robust accuracy, and (c) Difference in Robust accuracy between the last and best epochs are reported. ACAT has better performance and stability even at large training $\varepsilon$ values. Robust Accuracy is reported against GAMA attack [38] on $\varepsilon = 8/255$.

Curriculum learning approaches [37, 4] have been used to improve robustness-accuracy trade-off in adversarial training. Prior works [36, 1] also show that training convergence at large $\varepsilon$ bounds can be improved by linearly increasing $\varepsilon$ as training progresses. Inspired by this, we propose to linearly increase $\varepsilon$ alongside a cosine learning rate schedule and weight-space smoothing for improving the stability and convergence of training (Ref: Algorithm-2 in the Appendix). The use of a lower learning rate at larger $\varepsilon$ improves training stability. As shown in Fig.3 and Table-7 of the Appendix, the performance and stability of the ACAT are significantly better when compared to the TRADES-AWP 2-step baseline, specifically at larger perturbation bounds and higher model capacities, at the same computational cost. The proposed defense maintains a good clean accuracy at all the training $\varepsilon$ values considered, and has $< 0.1\%$ robustness difference between last and best epochs.

We thus show that even without modifying the training loss function it is possible to obtain significant gains in the stability and performance of 2-step adversarial training by using the proposed Ascending Constraint Adversarial Training. The proposed method can indeed improve the performance of other defenses such as GAT [38] and NuAT [39] as well, as shown in Table-8 of the Appendix.

**Analysis of the proposed ACAT defense:**

**Conjecture-2:** We hypothesize that given a Neural Network $f_{\theta_\varepsilon}$ that minimizes the TRADES loss [50] within a maximum perturbation radius of $\varepsilon$, and has sufficient smoothness in weight space within an $\ell_2$ radius $\psi$ around $\theta_\varepsilon$, there exist $\varepsilon' > \varepsilon$ and a model $f_{\theta_{\varepsilon'}}$ where $||\theta_{\varepsilon'} - \theta_\varepsilon|| \leq \psi$, such that $f_{\theta_{\varepsilon'}}$ has a lower TRADES loss within $\varepsilon'$ when compared to $f_{\theta_\varepsilon}$.

**Justification** presented in Appendix-E.3.

Based on Conjecture-2, by selecting a small enough learning rate $\eta$ and large enough $\psi$, it is possible to move towards a better solution w.r.t. the TRADES loss at $\varepsilon'$. The smoothness of loss surface further ensures that the gradients at $\varepsilon'$ are bounded with a low magnitude, preventing the onset of catastrophic overfitting, which is known to be associated with large magnitude gradients [26]. This motivates the proposed approach ACAT that effectively combines an increasing $\varepsilon$ schedule, cosine learning rate schedule that allows low learning rates at larger $\varepsilon$ values, and adversarial weight perturbation to ensure smoothness in the loss surface.

**Stability improvements obtained using ACAT:** We consider an extreme case of using single-step attacks for training, to demonstrate the stability improvements obtained using ACAT. We consider a checkpoint obtained at $\varepsilon = 7.25/255$ while training using the proposed ACAT defense on CIFAR-10 dataset for a WideResNet-34-10 architecture. We use this model as an initialization and perform single-step adversarial training using the TRADES-AWP training objective at $\varepsilon=7.5/255$. Our goal here is to merely verify the stability of single-step training, although the use of single-step attack makes the robustness gains marginal. We present the clean accuracy and robust accuracy against PGD-20 attack using different values of Adversarial Weight Perturbation step size $\gamma$ in Fig.4(a). It is evident from the plots that for a fixed value of learning rate, increasing $\gamma$ can delay the onset of gradient masking, thereby improving the training stability. We also note marginal improvements in performance at higher $\gamma$, due to the enhanced generalization. Therefore, weight-space smoothing indeed helps in improving the performance and stability of the proposed defense. We also note from Fig.4(b) that for a fixed value of $\gamma = 0.01$, lower learning rate leads to better stability, therefore cosine schedule also helps in improving stability. We note that the value of gamma in relation to the learning rate is crucial. Therefore, in the proposed approach, we use a fixed value of $\gamma$ and reduce the learning rate using a cosine learning rate schedule to improve training stability as $\varepsilon$ increases.

Table 2: **CIFAR-10:** Performance (%) of the proposed defenses ACAT and DAJAT when compared to the state-of-the-art. Robust evaluations are performed on GAMA [38] and AutoAttack [6]. Training time per epoch is reported by running each algorithm across 2 V100 GPUs.

| | | CIFAR-10, ResNet-18 | | | | CIFAR-10, WideResNet-34 | | |
|---|---|---|---|---|---|---|---|---|
| Method | Steps | Clean | GAMA | AutoAttack | Time/epoch (sec) | Clean | GAMA | AutoAttack |
| NuAT2-WA [39] | 2 | 82.21 | **50.97** | **50.75** | 109 | 86.32 | 55.08 | 54.76 |
| ACAT, Ours (Base, 2step) | 2 | **82.41** | 50.00 | 49.80 | 95 | **86.71** | **55.58** | **55.36** |
| PGD-AT [27] | 10 | 81.12 | 49.08 | 48.75 | 182 | 86.07 | 52.70 | 52.19 |
| TRADES-AWP [50, 44] | 10 | 80.47 | 50.06 | 49.87 | 228 | 85.19 | 55.87 | 55.69 |
| TRADES-AWP-WA | 10 | 80.41 | 49.89 | 49.67 | 228 | 85.10 | 56.07 | 55.87 |
| TRADES-AWP-WA (200 epochs) | 10 | 81.99 | 51.65 | 51.45 | 228 | 85.36 | 56.35 | 56.17 |
| DAJAT, Ours (Base, AA) | 2 + 2 | 85.60 | 51.27 | 51.06 | 160 | 87.87 | 56.97 | 56.68 |
| DAJAT, Ours (Base, 2*AA) | 2 + 4 | 85.99 | 51.71 | 51.48 | 219 | **88.90** | 57.22 | 56.96 |
| DAJAT, Ours (Base, 3*AA) | 2 + 6 | **86.67** | **51.81** | **51.56** | 280 | 88.64 | **57.34** | **57.05** |

Table 3: **CIFAR-100, ImageNette:** Performance (%) of the proposed defense DAJAT when compared to the state-of-the-art. Robust evaluations are performed on GAMA [38] and AutoAttack [6].

| | | CIFAR-100, ResNet-18 | | | CIFAR-100, WideResNet-34 | | | IN-10, ResNet-18 | | |
|---|---|---|---|---|---|---|---|---|---|---|
| Method | No. Steps | Clean | GAMA | AutoAttack | Clean | GAMA | AutoAttack | Clean | GAMA | AutoAttack |
| TRADES-AWP [50, 44] | 10 | 58.81 | 25.51 | 25.30 | 62.41 | 29.70 | 29.54 | 82.73 | 57.52 | 57.40 |
| TRADES-AWP-WA | 10 | 59.88 | 25.81 | 25.52 | 62.73 | 29.92 | 29.59 | 82.03 | 57.04 | 56.89 |
| ACAT, Ours (Base, 2step) | 2 | 62.05 | 26.35 | 26.10 | 65.75 | 30.61 | 30.23 | 82.34 | 57.12 | 56.96 |
| DAJAT, Ours (Base, AA) | 2 + 2 | 65.75 | 27.58 | 27.21 | 67.82 | 31.65 | 31.26 | 85.27 | 61.50 | 61.19 |
| DAJAT, Ours (Base, 2*AA) | 2 + 4 | 66.84 | 27.61 | 27.32 | 68.74 | 31.58 | **31.30** | 86.01 | **62.52** | **62.31** |
| DAJAT, Ours (Base, 3*AA) | 2 + 6 | **66.96** | **27.90** | **27.62** | **70.35** | 31.15 | 30.89 | **86.92** | 62.14 | 61.89 |

# 6 Experiments and Results

## 6.1 State-of-the-art comparison

We compare the performance of the proposed defenses ACAT and DAJAT against the 2-step and 10-step state-of-the-art defenses NuAT2-WA [38], TRADES-AWP [50, 44] and PGD-AT [27, 34] in Tables- 2, 3 and 4 on the CIFAR-10, CIFAR-100 [25] and ImageNette [20, 10]. We show in Table-16 of the Appendix that TRADES-AWP outperforms several other defenses in most settings, and the proposed defense DAJAT consistently outperforms all methods, showing substantial gains in performance. Further, since we integrate the proposed approach with TRADES-AWP, we primarily compare our approach with this across all datasets. We show in Table-17 of the Appendix that DAJAT can be integrated with other base defenses [1, 32] as well to obtain performance gains. We integrate model weight averaging with the TRADES-AWP defense (termed as TRADES-AWP-WA) as well for a fair comparison. We train all models for 110 epochs unless specified otherwise.

**ACAT:** We compare the performance of the proposed 2-step defense ACAT with the existing state-of-the-art 2-step defense NuAT2-WA [39] in the first partition of Table-2. While we achieve similar performance on ResNet-18, we obtain improvements in both clean and robust accuracy on WideResNet-34-10. We show in Section-E.4 that our proposed defense ACAT can be integrated with the Nuclear Norm training objective as well to obtain improved results. The performance of the proposed ACAT defense is superior to PGD-AT (10-step) defense [34] as well. When compared to the TRADES-AWP (10-step) defense, ACAT achieves substantial gains in clean accuracy and improved robust accuracy on CIFAR-100 (Table-3), and improved clean accuracy at a slight drop in robust accuracy on CIFAR-10 (Table-2), at half the computational cost.

**DAJAT:** We present three variants of the proposed defense DAJAT, by using one, two and three AutoAugment based augmentations for every image, denoted as (Base, AA), (Base, 2*AA) and (Base, 3*AA) respectively. From Tables-2 and 3, we note that even by using only a single augmentation (Base, AA), we obtain improved clean and robust accuracy when compared to most of the baselines considered across all datasets and models. By increasing the number of augmentations to 2 (Base, 2*AA), we observe consistent gains in robust and clean accuracy in all cases. In this setting, the computational complexity of the proposed approach matches with that of TRADES-AWP [44] as shown in Table-2. With the setting (Base, 3*AA), we further obtain marginal improvements in performance over (Base, 2*AA). Overall, using the (Base, 2*AA) approach, which has comparable time complexity as the TRADES-AWP 10-step defense, we obtain large gains ranging from 3.8% to 7% on clean accuracy and $\sim 1.8\%$ in robust accuracy against AutoAttack [6] across most settings. On the Imagenette dataset we obtain 4.2% higher clean accuracy and 4.49% higher robust accuracy, showing that augmentation strategies work best when the amount of training data is less when

Table 4: Performance (%) of DAJAT using **higher attack steps** and 200 training epochs on **CIFAR-10 and CIFAR-100**. Evaluations are performed against AutoAttack [6] and Common Corruptions (CC) [17].

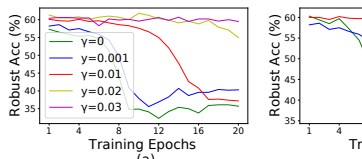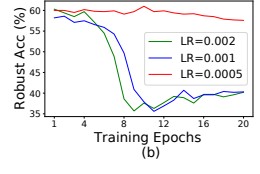

| Dataset | Model | Method | Clean | AutoAttack | CC |
|---------|-------|--------|-------|-----------|-----|
| CIFAR-10 | ResNet-18 | TRADES-AWP-WA | 81.99 | 51.45 | 72.64 |
| | | Ours (Base, 2*AA ) | **85.71** | **52.50** | **76.13** |
| | WRN-34-10 | TRADES-AWP-WA | 85.36 | 56.17 | 75.83 |
| | | Ours (Base, 2*AA ) | **88.71** | **57.81** | **80.12** |
| CIFAR-100 | ResNet-18 | TRADES-AWP-WA | 59.11 | 25.97 | 47.95 |
| | | Ours (Base, 2*AA ) | **65.45** | **27.69** | **54.85** |
| | WRN-34-10 | TRADES-AWP-WA | 60.30 | 28.68 | 48.95 |
| | | Ours (Base, 2*AA ) | **68.75** | **31.85** | **56.95** |

Figure 4: **Impact of (a) AWP step size $\gamma$ and (b) learning rate (LR) on stability of single-step TRADES-AWP training** using $\varepsilon = 7.5/255$ on a model pretrained upto $\varepsilon = 7.25/255$ using ACAT. (a) For a fixed LR, increasing $\gamma$ improves training stability. (b) For a fixed value of $\gamma = 0.01$, lower LR leads to better stability.

compared to the complexity of the task. We observe a similar trend using smaller subsets of training data on CIFAR-10 as shown in Fig.8 of the Appendix.

**Scaling to higher attack steps, longer training and larger models:** While the use of 2 attack steps helps in improving the training efficiency of DAJAT, we show that by using more attack steps and longer training epochs, we can indeed obtain further gains in performance. The varying $\varepsilon$ schedule in DAJAT allows the use of an increasing schedule in the number of steps as well, thereby limiting the overall cost associated with higher attack steps. We present results by increasing the number of attack steps from 2 to 5 uniformly every 50 epochs in Table-4 We obtain $\sim 3.5\%$ and $6 - 9\%$ higher clean accuracy, along with $1 - 1.6\%$ and $1.7 - 3.2\%$ gains in robust accuracy against AutoAttack on CIFAR-10 and CIFAR-100 respectively when compared to the TRADES-AWP-WA 200 epoch baseline. This achieves better robustness-accuracy trade-off when compared to all existing defenses that use the same model architecture on the RobustBench leaderboard [7]. On the CIFAR-100 RobustBench leaderboard, we outperform all existing methods using the same model architecture, w.r.t. both clean and robust accuracy, including the ones that use additional data, showing that data augmentations can be effectively used to overcome the large sample complexity of adversarial training. The use of diverse augmentations in DAJAT also improves the generalization of the model to Common Corruptions by 4% and 8% on CIFAR-10-C and CIFAR-100-C datasets [17] respectively as shown in Table-4. Further, we find that the performance gains obtained using DAJAT are higher on larger capacity models (Ref: Table-10 of the Appendix).

**Computational Efficiency:** As shown in Appendix-F.6, the use of ACAT strategy in the proposed DAJAT defense enables us to achieve similar computational complexity as TRADES-AWP defense, while obtaining gains in performance. Using DAJAT, we achieve 10% reduction in FLOPs (training) and training time, while obtaining 3.8-5.5% higher clean accuracy and 1-1.6% higher robust accuracy.

### 6.2 Ablation experiments

We present ablation experiments to highlight the significance of different components of the proposed approach in Table-5 on the CIFAR-10 dataset using ResNet-18 architecture. All experiments are run for 110 training epochs, except A7 which is run for 220 epochs. We show the importance of the JS divergence term in the proposed loss in the ablations A1-A6 of Table-5. Using the JS divergence term we obtain $\sim 1\%$ higher clean accuracy across (Base, AA), (Base, 2*AA) and (Base, 3*AA) settings of the proposed defense. For (Base, 2*AA) and (Base, 3*AA) we obtain marginal improvements in robust accuracy as well. From A7, A9 and A10, we find that the proposed JS divergence term helps even in the case where both augmentations of an image are generated using the same pipeline. Using two AutoAugment based transformations, we obtain 1.6% higher clean accuracy when compared to the 220 epoch 2-step defense at a comparable computational cost. Comparing A9, A10 and A11, we note that the use of simple and complex augmentations indeed shows improvements over the case of using 2 complex or simple augmentations alone. The importance of split-batch norm in the proposed approach can be evidently seen by comparing A12 and A14. By using single batch norm (A12), robust accuracy drops by 8.24%. Further, in this case if the JS term is also dropped, the robustness of the network is almost completely lost. This shows that using a single batch norm layer for diverse augmentations makes it harder for the network to converge. We also note that the JS divergence term indeed helps in improving the convergence of training in addition to improving performance. From Appendix-F.4, we note that the use of varying $\varepsilon$ schedule in DAJAT improves the stability of training resulting in large performance gains w.r.t. the use of fixed $\varepsilon$, specifically at larger model capacities.

Table 5: **Ablation experiments** performed on the **CIFAR-10** dataset using ResNet-18 architecture. Robust Accuracy is reported against GAMA attack [38]

| Method | # Steps | Clean | Robust | Method | # Steps | Clean | Robust |
|---|---|---|---|---|---|---|---|
| [A1] Ours (Base, AA ), no JS div | 2 + 2 | 84.55 | 51.45 | [A8] Ours (Base, 2step) | 2 | 82.41 | 50.00 |
| [A2] Ours (Base, AA ) | 2 + 2 | 85.60 | 51.27 | [A9] Ours (AA, AA ) | 2 + 2 | 84.68 | 49.71 |
| [A3] Ours (Base, 2*AA ), no JS div | 2 + 4 | 85.07 | 51.53 | [A10] Ours (Base, Base ) | 2 + 2 | 83.93 | 49.88 |
| [A4] Ours (Base, 2*AA ) | 2 + 4 | 85.99 | 51.71 | [A11] Ours (Base, AA ) | 2 + 2 | 85.60 | 51.27 |
| [A5] Ours (Base, 3*AA ), no JS div | 2 + 6 | 85.31 | 51.67 | [A12] Ours (Base, 3*AA ) Single BN | 2 + 6 | 86.68 | 43.57 |
| [A6] Ours (Base, 3*AA ) | 2 + 6 | 86.67 | 51.81 | [A13] Ours (Base, 3*AA ) Single BN, no JS | 2 + 6 | 75.64 | 4.20 |
| [A7] Ours (Base, 2step), 220 epochs | 2 | 83.05 | 50.31 | [A14] Ours (Base, 3*AA ) | 2 + 6 | 86.67 | 51.81 |

Table 6: Impact of using **other augmentations** in DAJAT. Performance on CIFAR-10 dataset with ResNet-18 architecture is reported. Robust evaluations are done against GAMA attack [38]. [†]PreAct-ResNet18 with Swish activation is used [33, 31].

| Augmentation | Augmentation | | Base + Aug | | Base + 2 * (Aug) | | Augmentation | Augmentation | | Base + Aug | | Base + 2 * (Aug) | |
|---|---|---|---|---|---|---|---|---|---|---|---|---|---|
| | Clean | Robust | Clean | Robust | Clean | Robust | Augmentation | Clean | Robust | Clean | Robust | Clean | Robust |
| No Augmentation | 76.32 | 43.20 | 78.08 | 41.71 | 77.42 | 41.07 | Cutout [11] | 82.38 | **50.14** | 84.91 | **51.40** | 85.11 | 51.60 |
| Pad+Crop+H-Flip | 82.41 | 50.00 | 83.69 | 51.30 | 83.62 | 51.09 | Colour Jitter | **82.98** | 48.82 | 84.50 | 51.19 | 84.85 | 51.62 |
| AutoAugment [8] | 82.54 | 48.11 | 84.94 | 51.23 | **85.99** | **51.71** | Mixup [49] | 79.08 | 45.07 | 85.18 | 50.18 | 84.24 | 50.01 |
| Cutmix [48] | 79.03 | 41.57 | 82.33 | 50.90 | 81.64 | 49.50 | RandAugment [9] | 82.48 | 44.66 | 84.61 | 51.01 | 85.47 | 51.33 |
| Cutmix[†] [33, 31] | 82.01 | 47.65 | 84.58 | 50.97 | 85.49 | 51.58 | Augmix [18] | 82.38 | 48.84 | **84.96** | 50.4 | 85.18 | 50.51 |

## 6.3 Combining the proposed defense with other augmentations

We explore combining the proposed defense DAJAT with other augmentations in Table-6, and present the impact of individual augmentations in AutoAugment in Table-4 of the Appendix. We do not use the JS divergence term for Cutmix and Mixup since they involve changes in the label space. Without using any augmentation in the training dataset, we obtain poor clean and robust accuracy, highlighting the importance of the simple augmentations. The proposed approach obtains good performance gains using AutoAugment [8], Color Jitter and CutOut [11] augmentations, highlighting that it can work well with pixel-level and spatial augmentations. In case of CutMix [48] and Mixup [49], since the base accuracy obtained by using the augmentation alone is itself very poor, combining them with the proposed approach does not yield competent results. However they are considerably better than naively using the augmentation. This shows that the proposed approach enables the use of a variety of augmentations without the need for careful selection. The use of Cutmix and Mixup in adversarial training are challenging since they involve changes in label space. We note that Rebuffi et al. [33] obtain considerable gains using Cutmix along with many other improvements. However, naively augmenting using CutMix leads to a considerably degraded performance. By incorporating some of the tricks reproduced by Rade et al. [31] we obtain improvements in the CutMix baseline and obtain further gains in the proposed method as well (Ref: Appendix-F.7, Table-15 of the Appendix).

## 7 Conclusion

Contrary to prior knowledge, we show that it is indeed possible to use common augmentation strategies that modify the low-level statistics of images to improve the performance of adversarial training. We propose a novel defense Diverse Augmentation based Joint Adversarial Training (DAJAT), that uses a combination of simple and complex augmentations with separate batch normalization layers to allow the network training to benefit from the diverse training data distribution obtained using complex augmentations, while also specializing on a distribution that is close to the test set. The use of JS divergence term between network predictions of different augmentations enables the joint learning across various augmentations. We improve the efficiency of the proposed defense by utilizing the proposed approach Ascending Constraint Adversarial Training (ACAT), that improves the stability and performance of TRADES 2-step adversarial training significantly by using a linearly increasing $\varepsilon$ schedule along with a cosine learning rate schedule and weight-space smoothing.

**Limitations:** While we focus our work on *how* to use augmentations effectively in adversarial training, we do not focus on *which* augmentations are best suited for the same. We believe this work can open up further possibilities towards finding better data augmentations for adversarial training.

## 8 Acknowledgments and Disclosure of Funding

This work was supported by a research grant (CRG/2021/005925) from SERB, DST, Govt. of India. Sravanti Addepalli is supported by a Google PhD Fellowship in Machine Learning.

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
