# Supplementary Material: Efficient and Effective Augmentation Strategy for Adversarial Training

**Sravanti Addepalli**[†][*]     **Samyak Jain**[†][◇][‡][*]     **R.Venkatesh Babu**[†]

[†] Video Analytics Lab, Indian Institute of Science, Bangalore

[◇] Indian Institute of Technology (BHU) Varanasi

## A    Details on the proposed defense DAJAT

The algorithm of the proposed approach is presented in Algorithm-1. In every training iteration, multiple augmentations are considered for every image $x_i$ (L7). We consider one base augmentation and $T$ complex augmentations. The base augmentation consists of Pad and Crop followed by Horizontal Flip, while the complex augmentations are a combination of AutoAugment [7] and the base augmentations. The attack generation for each augmentation (L8-L13) is similar to the ACAT algorithm discussed in Section-E.2. The DAJAT loss (L16) is a combination of the TRADES loss [25] (L17) on each augmentation, and a Jensen-Shannon (JS) divergence term between all augmentations. The JS divergence is a combination of KL divergence terms with respect to the average probability vector as shown below.

$$\text{JSD}(f_\theta(x_{i,\text{base}}), f_\theta(x_{i,\text{auto}(1)}), \ldots, f_\theta(x_{i,\text{auto}(T)})) = \frac{1}{T+1}\big\{\text{KL}(f_\theta(x_{i,\text{base}}, M)) +$$
$$\text{KL}(f_\theta(x_{i,\text{auto}(1)}, M)) + \cdots + \text{KL}(f_\theta(x_{i,\text{auto}(T)}, M))\big\} \quad (1)$$

where $M$ is defined as below,

$$M = \frac{1}{T+1}\big\{f_\theta(x_{i,\text{base}}) + f_\theta(x_{i,\text{auto}(1)}) + \cdots + f_\theta(x_{i,\text{auto}(T)})\big\} \quad (2)$$

As shown in Table-5 of the main paper, the JS-divergence term improves accuracy on clean samples and training convergence by enabling the joint learning of representations across different augmentations. The model weights are perturbed by $\tilde{\theta}$, by maximizing the TRADES loss on the base augmentations alone within the constraint set $\mathcal{M}(\theta)$ (L18). This constraint set is chosen such that $||\tilde{\theta}_l|| \leq \gamma \cdot ||\theta_l||$ for any layer $l$. The network at $\theta$ is then updated using gradients at $f_{\theta+\tilde{\theta}}$ to minimize the overall loss $\mathcal{L}_{\text{DAJAT}}(\theta + \tilde{\theta})$ (L20).

## B    Augmentations

While existing works hypothesize that augmentations changing the low-level statistics of images cannot effectively improve adversarial training [18], we show in this work that with the use of split Batch-normalization layers and JS divergence term between different augmentations, it is indeed possible to obtain significant gains using augmentations that modify the low-level statistics of images as well. In this work, we use an existing augmentation strategy, AutoAugment to obtain an improvement in performance using the proposed training algorithm DAJAT. AutoAugment uses Proximal Policy Optimization to find the set of policies that can yield optimal performance on a given dataset. It consists of 25 unique sub-policies for a given dataset, where each sub-policy

---

[*]Equal Contribution.

Correspondence to Sravanti Addepalli <sravantia@iisc.ac.in>, Samyak Jain <samyakjain.cse18@itbhu.ac.in>

[‡] Work done during internship at Video Analytics Lab, Indian Institute of Science.

36th Conference on Neural Information Processing Systems (NeurIPS 2022).

**Algorithm 1** Diverse Augmentation based Joint Adversarial Training (DAJAT)

---

1: **Input:** Network $f_\theta$, Training Dataset $\mathcal{D} = \{(x_i, y_i)\}$, Adversarial Threat model: $\ell_\infty$ bound of radius $\varepsilon$, number of epochs E, Maximum Learning Rate $\text{LR}_{max}$, $M$ training mini-batches of size $n$, number of attack steps $S$, Cross-entropy loss $\ell_{CE}$, Weight perturbation constraint $\mathcal{M}(\theta)$, Number of augmented images using autoaugment $T$, coefficient of KL divergence term $\beta$

2: **for** $epoch = 1$ **to** $E$ **do**

3:    $\varepsilon_{\text{asc}} = epoch \cdot \varepsilon/E$

4:    $\text{LR} = 0.5 \cdot \text{LR}_{max} \cdot (1 + cosine((epoch - 1)/E \cdot \pi))$

5:    **for** $iter = 1$ **to** $M$ **do**

6:      **for** $i = 1$ **to** $n$ (in parallel) **do**

7:        **for** $a \in \{\text{base}, \text{auto}(1), \ldots, \text{auto}(\text{T})\}$ **do**

8:          **for** $steps = 1$ **to** $S$ **do**

9:            $\delta = 0.001 \cdot \mathcal{N}(0, 1)$

10:            $\delta = \delta + \varepsilon_{\text{asc}} \cdot \text{sign}\left(\nabla_\delta \text{KL}(f_\theta(x_{i,a})||f_\theta(x_{i,a} + \delta))\right)$

11:            $\delta = Clamp\left(\delta, -\varepsilon_{\text{asc}}, \varepsilon_{\text{asc}}\right)$

12:            $\widetilde{x}_{i,a} = Clamp\left(x_{i,a} + \delta, 0, 1\right)$

13:          **end for**

14:        **end for**

15:      **end for**

16:
$$\mathcal{L}_{\text{DAJAT}}(\theta) = \frac{1}{T+1} \cdot \frac{1}{n} \sum_{i=1}^{n} \left\{ \mathcal{L}_{\text{TR}}(\theta, (x_i, \tilde{x}_i)_{\text{base}}, y_i) + \sum_{t=1}^{T} \mathcal{L}_{\text{TR}}(\theta, (x_i, \tilde{x}_i)_{\text{auto}}, y_i) \right\}$$
$$+ \frac{1}{n} \sum_{i=1}^{n} \left\{ \text{JSD}(f_\theta(x_{i,\text{base}}), f_\theta(x_{i,\text{auto}(1)}), \ldots, f_\theta(x_{i,\text{auto}(\text{T})})) \right\}$$

17:      where, $\mathcal{L}_{\text{TR}}(\theta, (x, \tilde{x}), y) = \mathcal{L}_{\text{CE}}(f_\theta(x), y) + \beta \cdot \text{KL}(f_\theta(x)||f_\theta(\tilde{x}))$

18:      $\tilde{\theta} = \underset{\hat{\theta} \in \mathcal{M}(\theta)}{\text{argmax}} \frac{1}{n} \sum_{i=1}^{n} \left\{ \mathcal{L}_{\text{TR}}(\theta + \hat{\theta}, (x_i, \tilde{x}_i)_{\text{base}}, y_i) \right\}$

19:      $\theta = \theta - \text{LR} \cdot \nabla_\theta(\mathcal{L}_{\text{DAJAT}}(\theta + \tilde{\theta}))$

20:    **end for**

21: **end for**

---

is a combination of two augmentations chosen from a set of pre-defined augmentations in series. The pre-defined augmentations include the spatial transformations - shear, rotation and translation, and augmentations that cause changes in low-level statistics of images - color, posterize, solarize, brightness, contrast, sharpness, autocontrast, equalize and invert.

We visualize some of the augmentations generated using AutoAugment and Base augmentations (Pad+Crop, Horizontal Flip) in Fig.1 and Fig.2 respectively. It can be noted that multiple augmentations generated using AutoAugment are significantly more diverse than the augmentations generated using the base augmentations. The use of multiple diverse augmentations leads to improved generalization on test set as discussed in Section-4 of the main paper. Further we visualize the perturbations generated using the 2-step attack with KL-divergence loss on CIFAR-10 images without augmentations, with Base augmentations and with AutoAugment on Trades-AWP [23] model trained at $\varepsilon = 8/255$ in Fig.3, Fig.4 and Fig.5 respectively. For plotting, one image is selected at random from each of the ten classes. Because of the increased diversity amongst the images on using AutoAugment, we observe more diversity in the perturbations as well (Fig.5) when compared to the Base augmentations (Pad+Crop, Horizontal Flip) (Fig.4) and No augmentations (Fig.3) case.

### B.1 Distinction between Simple and Complex Augmentations

We term the augmentations that preserve low-level features of images as simple augmentations, and those that modify the same as complex augmentations. To distinguish between simple and complex augmentations, we do not use the difference between two images in pixel-space, since this would incorrectly show that simple changes like horizontal-flip and crop are far apart. Instead, we use metrics that better capture low-level features at pixel and patch levels. This can be measured at a pixel-level using MSE between color histograms, and at a patch-level using patch-wise MSE. To

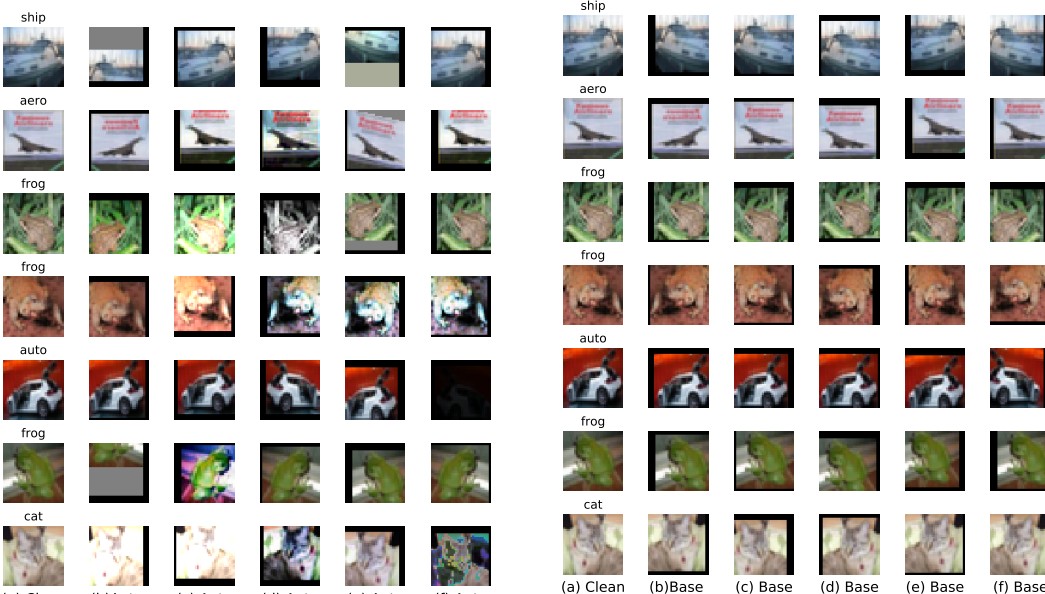

Figure 1: CIFAR-10 images in (a), along with respective random augmentations generated using AutoAugment [7] shown in columns (b-f)

Figure 2: CIFAR-10 images in (a), along with respective random augmentations generated using Base augmentations (Pad+Crop and Horizontal Flip) shown in columns (b-f)

compute patch-wise MSE between two images $x_1$ and $x_2$, for every $8 \times 8$ patch in $x_1$ we find the nearest patch in $x_2$ and a horizontal flip of $x_2$, and compute an average across all patches in $x_1$. We report the mean and standard deviation of this value across all images in the test set. We show the pair-wise distances between three sets of images (Unaugmented, Pad+Crop+HFlip, AutoAugment) in Table-1. Lower value indicates that the images are more similar. We note that Pad+Crop+HFlip augmentations have the advantage of being more similar to the distribution of unaugmented images that are expected during inference, while AutoAugment transformed images are farther away from the unaugmented images. AutoAugment consists of several augmentations of varying complexity levels, and may contain augmentations of similar complexity as the Base augmentations (Pad+Crop+HFlip) as well. This is reflected in the higher variance in pair-wise distances corresponding to AutoAugment.

Table 1: Distinction between Simple and Complex Augmentations in pixel space, in terms of MSE between Histograms and MSE between Patches.

| Image pairs | Complexity | MSE between Histograms | MSE between Patches |
|---|---|---|---|
| Base (Pad+Crop+HFlip), Unaugmented | Simple | $133.60_{\pm 94.05}$ | $43.68_{\pm 23.37}$ |
| AutoAugment, Unaugmented | Complex | $289.25_{\pm 405.11}$ | $51.39_{\pm 24.23}$ |

DAJAT allows separate function mappings for augmentations that resemble the inference-time distribution (Pad+Crop+HFlip), and those that lead to better diversity (AutoAugment). Base augmentations have low variation, and are similar to the distribution of unaugmented images, which is important to obtain performance gains using the batch-norm layer corresponding to these base augmentations during inference. On the other hand, the high variance of AutoAugment based transformations helps in improving the robust generalization of the overall model.

## B.2   Distribution shift of Natural and Adversarial images due to Augmentations

We compare the distribution shift between (natural-augmented, natural-unaugmented) pairs and (adversarial-augmented, adversarial-unaugmented) pairs. We consider two types of distances between image pairs: low-level (MSE between histograms/patches) and feature-level (FID). In terms of low-level distances, we expect the distances between (natural-augmented, natural-unaugmented) pairs and the corresponding (adversarial-augmented, adversarial-unaugmented) pairs to be similar, since

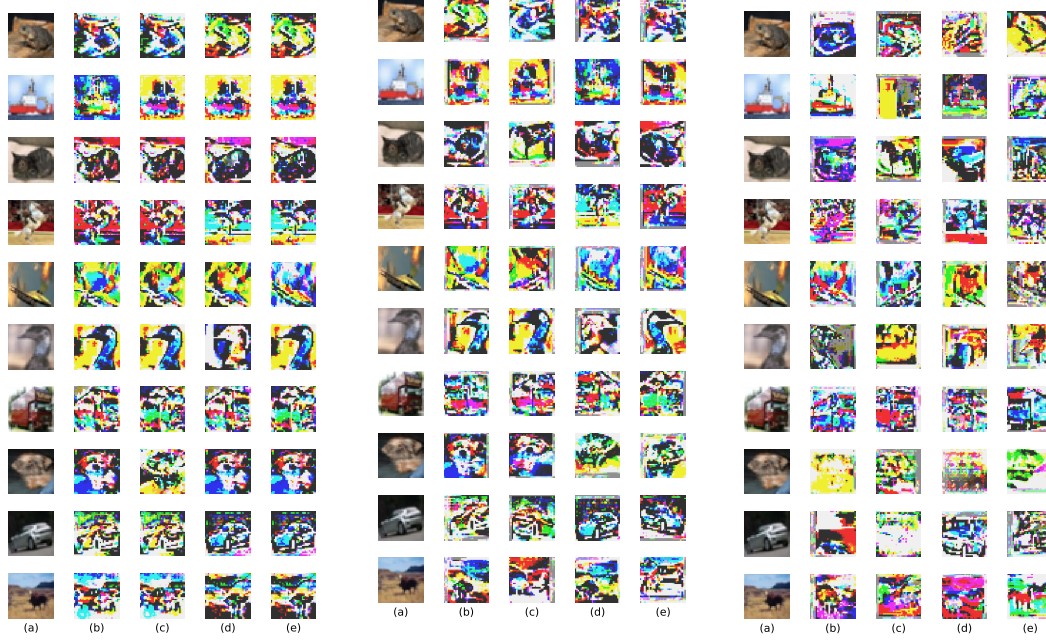

Figure 3: CIFAR-10 images in (a), along with perturbations of respective images generated using without any augmentation shown in columns (b-e). The attack is generated at $\varepsilon = 8/255$ and the corresponding perturbation's magnitude is scaled up for better visibility

Figure 4: CIFAR-10 images in (a), along with perturbations of respective random augmentations generated using Base augmentations (Pad+Crop and Horizontal Flip) shown in columns (b-e). The attack is generated at $\varepsilon = 8/255$ and the corresponding perturbation's magnitude is scaled up for better visibility

Figure 5: CIFAR-10 images in (a), along with perturbations of respective random augmentations generated using AutoAugment [7] shown in columns (b-e).The attack is generated at $\varepsilon = 8/255$ and the corresponding perturbation's magnitude is scaled up for better visibility

the perturbed images are only an $\varepsilon$ away from natural images. However, as seen in Fig.3, 4 and 5, the perturbations of Pad+Crop+HFlip look similar to the perturbations of unaugmented images, while the perturbations of AutoAugment based images look different from those of unaugmented images. This is a result of larger pixel-level differences between the (natural-AutoAugment, natural-unaugmented) image pairs when compared to (natural-PadCrop, natural-unaugmented) image pairs, which serves as a more diverse initialization for the attack. The difference in the absolute perturbations results in a larger distance in feature space (Fréchet Inception Distance or FID) as shown in Table-2.

Table 2: FID between Augmented and Unaugmented images with Simple and Complex augmentations

| Image pairs | FID between Natural image pairs | FID between Adversarial image pairs |
|---|---|---|
| Base (Pad+Crop+HFlip), Unaugmented | 24.02 | 33.41 |
| AutoAugment, Unaugmented | 37.62 | 43.75 |

For better clarity, we summarize our findings in Table-3. The higher feature level distance between (adversarial-AutoAugment, adversarial-unaugmented) image pairs when compared to (natural-AutoAugment, natural-unaugmented) image pairs translates to higher $\frac{1}{2}d_{\mathcal{F}\Delta\mathcal{F}}(s,t)$ in Eq.1 of the main paper. Based on Conjecture-1(ii), unless this difference is accounted for, complex augmentations cannot improve the performance of adversarial training.

## B.3 Justification on the choice of Simple and Complex Augmentation pipeline

In the proposed method, we use two sets of augmentations - Pad+Crop+HFlip and AutoAugment. The second set of augmentations (complex augmentations) consists of an autoaugment based transformation followed by the base augmentations (Pad+Crop+HFlip). This has two implications -

Table 3: Summary of pixel-level and feature-level distances between Augmented and Unaugmented image pairs for Natural and Adversarial Images. Base refers to the augmentations Pad+Crop+HFlip.

| Natural/ Adversarial | Distributions | Low-level distance | Feature-level distance |
|---|---|---|---|
| Natural images | Base, Unaugmented
Autoaugment, Unaugmented | Low
High | Low
Medium |
| Adversarial images | Base, Unaugmented
Autoaugment, Unaugmented | Low
High | Medium
**High** |

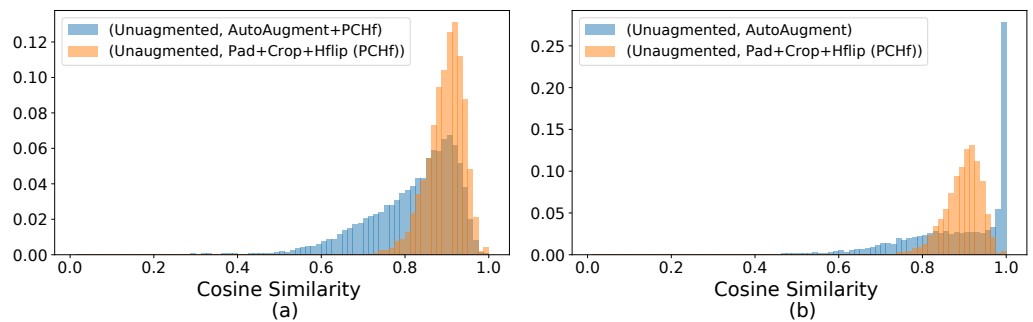

Figure 6: Normalized histogram of pair-wise cosine similarity between the features obtained using a pretrained Inception-V3 [21] model for different augmentations w.r.t. the respective input images. Histograms are plotted for the following pairs, (a) (Unaugmented, AutoAugment+Pad+Crop+Horizontal flip) and (Unaugmented, Pad+Crop+Horizontal flip), (b) (Unaugmented, AutoAugment) and (Unaugmented, Pad+Crop+Horizontal flip). The full test set of CIFAR-10 (10k images) is used for the plots.

firstly, this ensures that the complexity of these augmentations is always greater than or equal to the base augmentations. Secondly, since AutoAugment returns the unaugmented image as well, with a certain probability (0.22 for CIFAR-10 policy), the base augmentations form a subset of the complex augmentations. This trend is indeed reflected in the distribution of pair-wise feature-level similiarities (Cosine similarity between features obtained from an Inception-V3 network) between the following pairs: (Unaugmented, Pad+Crop+HFlip (PCHf)) and (Unaugmented, AutoAugment+PCHf) as shown in Fig.6(a).

When AutoAugment alone is applied, a large fraction of images have a very high cosine similarity, while others have a more spread out distribution as shown in Fig.6(b). When Pad+Crop+HFlip is applied in series after AutoAugment, the distribution of these images shifts to the left, leading to an overlap in the two distributions. Even with such a large overlap the method works well because, the role of the "complex" batch-norm layer is to allow the learning of a function that minimizes empirical risk across a wide distribution of data. While the test distribution may be different from these augmentations, learning from diverse data is known to prevent overfitting and improve generalization. However, since the task of adversarial training is inherently hard, and the objective of minimizing loss on a wider distribution of data makes the task harder, we observe a drop in overall accuracy. The use of a separate batch-norm layer for "simple" augmentations allows the network to specialize on a select subset that is close to the distribution of test set images, and has a low variance. While the diversity of simple augmentations is low, it is sufficient to learn the batch-norm statistics and affine parameters which constitute 0.05% of all parameters, while majority of the parameters are learned using both distributions, resulting in low overfitting.

## B.4 Ablations

We show the impact of some of the important categories of augmentations individually in Table-4.

The robust accuracy improves when the spatial augmentations - shear, rotation and translation are not used for adversarial training. Amongst the augmentations that modify the low-level statistics of images, change in color balance gives maximum benefit. Further, it can be noted that although some augmentations such as change in brightness and contrast lead to a drop in robust accuracy when used

Table 4: Impact of individual augmentations within AutoAugment [7] on the proposed defense DAJAT. Performance on CIFAR-10 dataset with ResNet-18 architecture is reported. Robust evaluations are done against GAMA attack [19]

| Augmentation | Augmentation | | Base + Augmentation + JS | |
|---|---|---|---|---|
| | Clean Acc | Robust Acc | Clean Acc | Robust Acc |
| AutoAugment | 82.54 | 48.11 | **84.94** | 51.23 |
| AutoAugment (without spatial augs) | **83.70** | 48.80 | 84.94 | 51.40 |
| Brightness | 82.11 | 46.42 | 84.56 | 50.94 |
| Sharpness | 81.78 | 49.85 | 84.30 | 50.54 |
| Color-Balance | 82.31 | **49.87** | 84.39 | **51.48** |
| Contrast | 82.45 | 46.54 | 84.21 | 50.77 |

Table 5: **Split Batch-Normalization:** Impact of using common/ split running statistics and affine parameters. Using a combination of separate running statistics and affine parameters works the best.

| Method | Clean Accuracy | Robust Accuracy |
|---|---|---|
| [E1] split running statistics + split affine parameters (Ours) | **88.90** | **57.22** |
| [E2] split running statistics + common affine parameters | 88.61 | 56.91 |
| [E3] common running statistics + split affine parameters | 88.86 | 57.01 |
| [E4] common running statistics + common affine parameters (single Batch-Norm) | 89.08 | 53.86 |

directly, the use of the same augmentations in the proposed framework results in a significant boost in accuracy. Therefore there is no need to use carefully selected augmentations with the proposed framework.

## C   Split Batch-Normalization

We perform three ablation experiments - first by training with separate running statistics and common affine parameters, second by training with common running statistics and separate affine parameters, and third without using split batch-norm at all, as shown in Table-5. Robust Accuracy is reported against the GAMA attack. As shown in the Table-5, E2 and E3 (having either split running statistics or split affine parameters) perform similar to the proposed approach, where separate running statistics and affine parameters are used. This shows that our method learns a different function mapping for both augmentations, and this can be realized by having different running statistics or different affine parameters or using a combination of both. We further note that the use of a single batch-norm layer for both augmentations (E4) degrades the results significantly.

We compare the average cosine similarity of the running statistics across training iterations for both our method, and the case where we have separate running statistics and common affine parameters in Fig.7. We note that the scale and trend of the average cosine similarity is similar in both cases. In the case where common affine parameters are used, the gains in results w.r.t. single batch-norm case can be attributed to the small drop in cosine similarity over training. This indicates that small changes in these running statistics can indeed lead to a large impact in the overall results (as shown in Table-5). We further verify this by noting the large difference in performance of the models when different batch-norm parameters are used for training, in Table-6. Robust Accuracy is reported against the GAMA attack.

## D   Improvements of DAJAT in the low data regime

We compare the performance of the proposed DAJAT defense (using Base, 2*AA) with TRADES-AWP [23] 2-step baseline across different sizes of the CIFAR-10 training dataset on ResNet-18 architecture in Fig.8. We consider class balanced dataset for each case. We note that across different settings, the proposed approach achieves 5-9% gains in clean accuracy and an average gain of 1.3% in robust accuracy. The gain in clean accuracy in the low data regime is significantly high (8.93%) highlighting the need for augmentations in improving Adversarial Training performance, specifically in real-world settings where the training data is low.

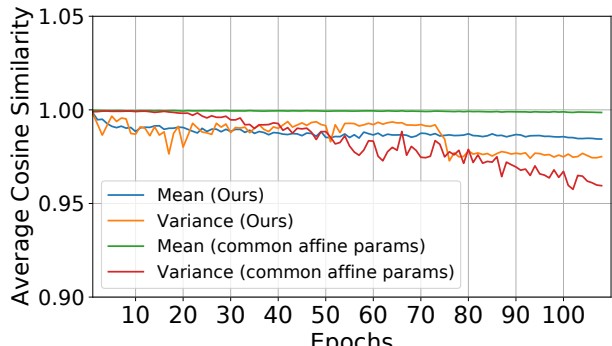

Figure 7: **Average Cosine Similarity (averaged over all the layers of WideResNet-34-10) of Running Mean and Variance** across training epochs. For the proposed approach DAJAT, and an ablation experiment that uses common batch-normalization affine parameters, the scale of cosine similarities are similar, indicating that small changes in running mean and variance can indeed have a large impact on model outputs.

Table 6: **Impact of using different batch-normalization layers** (Pad+Crop+HFlip vs AutoAugment) **at inference time.**

| Method | Batch-norm (Inference) | Clean Acc | Robust Acc |
|---|---|---|---|
| Split running statistics + split affine parameters (Ours) | Pad+Crop+HFlip | **88.90** | **57.22** |
| Split running statistics + split affine parameters (Ours) | AutoAugment | 76.17 | 44.45 |
| Split running statistics + common affine parameters | Pad+Crop+HFlip | 88.61 | 56.91 |
| Split running statistics + common affine parameters | AutoAugment | 78.69 | 45.41 |

# E    Details on ACAT

## E.1    Motivation: Instability of 2-step Adversarial Training

Ascending perturbation radius helps in mitigating gradient masking when lesser steps are used for attack generation. We present results of ACAT when compared to Fixed constraint AT (TRADES-AWP with 2 attack steps) on CIFAR-10 dataset using $\varepsilon = 8/255$ with ResNet-18 and WideResNet-34-10 model architectures in Table-7. Robust Accuracy is reported against the GAMA attack. Best accuracy is computed using PGD-20 attack, which is not very reliable. Hence, in some cases, best epoch may have a slightly lower accuracy when compared to the last epoch. On ResNet-18, we observe that the difference between last and best epochs for both methods is very low. However, on WideResNet-34-10, we observe the phenomenon of gradient masking in Fixed Constraint AT, with

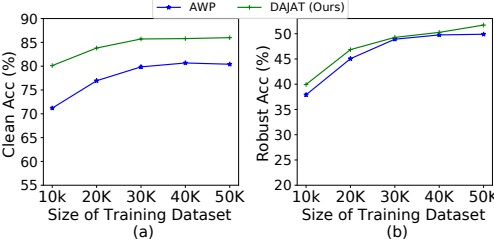

Figure 8: **Improvements of DAJAT in the low data regime:** Performance of proposed approach DAJAT (Base, 2*AA) when compared to the TRADES-AWP [23] 2-step baseline on ResNet-18 architecture and CIFAR-10 dataset. The proposed approach achieves improvements across different sizes of training dataset, with higher gains in the low data regime.

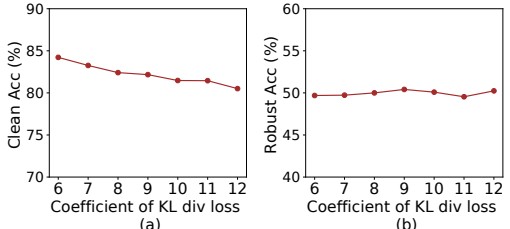

Figure 9: **Performance of ACAT across variation in** $\beta$**:** Performance of the proposed defense ACAT against variation in the coefficient of KL divergence loss between the clean and adversarial samples $\beta$. Higher $\beta$ leads to improved robust accuracy (against GAMA [19] attack) at the cost of clean accuracy. ResNet-18 architecture and CIFAR-10 dataset are used.

Table 7: Fixed vs ascending $\varepsilon$ Adversarial training on ResNet-18 and WRN-34-10 models with maximum $\varepsilon = 8/255$ on CIFAR-10 dataset with 2 step attack generation during training. Impact of gradient masking is higher for larger capacity models. This is effectively mitigated with the use of Ascending $\varepsilon$ Adversarial Training

| Architecture | Method | Clean(last) | Robust(last) | Clean(best) | Robust(best) | Clean(last - best) | Robust(last - best) |
|---|---|---|---|---|---|---|---|
| ResNet-18 | Fixed $\varepsilon$ AT | 80.63 | 49.63 | 80.82 | 49.61 | -0.19 | 0.02 |
| ResNet-18 | Ascending $\varepsilon$ AT | **82.41** | **50.00** | **82.57** | **49.91** | -0.16 | 0.09 |
| WRN-34-10 | Fixed $\varepsilon$ AT | 86.69 | 44.87 | **86.83** | 54.76 | -0.14 | -9.89 |
| WRN-34-10 | Ascending $\varepsilon$ AT | **86.71** | **55.58** | 86.30 | **55.46** | 0.41 | 0.12 |

robust accuracy dropping by around 10% towards the end of training. We note that the difference between last and best epochs is very low for ACAT even on WideResNet-34-10. The main motivation of using an ascending perturbation radius is to stabilize training and prevent the onset of gradient masking.

## E.2 ACAT training algorithm

The algorithm for the proposed ACAT defense is presented in Algorithm-2. We consider an $\ell_\infty$ threat model of perturbation radius $\varepsilon$. The perturbation bound for attack generation $\varepsilon_{\text{asc}}$ is linearly increased from 0 to $\varepsilon$ during training (L3). The learning rate follows a cosine schedule across the training epochs as shown in L4. The attack generation (L7-L12) is done for 2 iterations and follows the TRADES [25] settings. Initially Gaussian noise of magnitude 0.001 is added to every pixel. The KL divergence loss between the clean and perturbed images is maximized using the perturbation step size $\varepsilon_{\text{asc}}$. Further, the perturbation is clipped to remain within the threat model in every iteration.

As shown in L14, the TRADES-AWP [23] loss is used for adversarial training. The loss $\mathcal{L}_{\text{TR}}(\theta + \hat{\theta})$ is maximized with respect to $\hat{\theta}$ to find the perturbation $\tilde{\theta}$ within the constraint set $\mathcal{M}(\theta)$ (L15). Further, the model weights $\theta$ are update using gradients at $f_{\theta + \tilde{\theta}}$ (L16). The defense ACAT does not use any additional training hyperparameters when compared to the TRADES-AWP defense. We vary the hyperparameter $\beta$ to obtain optimal results. We show the trend of model performance across variation in $\beta$ in Section-F.2.

## E.3 Analysis of the proposed ACAT defense

**Conjecture-2:** We hypothesize that given a Neural Network $f_{\theta_\varepsilon}$ that minimizes the TRADES loss [25] within a maximum perturbation radius of $\varepsilon$, and has sufficient smoothness in weight space within an $\ell_2$ radius $\psi$ around $\theta_\varepsilon$, there exist $\varepsilon' > \varepsilon$ and a model $f_{\theta_{\varepsilon'}}$, where $||\theta_{\varepsilon'} - \theta_\varepsilon|| \leq \psi$, such that $f_{\theta_{\varepsilon'}}$ has a lower TRADES loss within $\varepsilon'$ when compared to $f_{\theta_\varepsilon}$.

**Justification:** We first consider Adversarial Training within a perturbation bound $\varepsilon$ using the TRADES-AWP loss shown below.

$$\mathcal{L}_{\text{AWP},\theta,\varepsilon} = \max_{\hat{\theta} \in \mathcal{M}(\theta)} \frac{1}{N} \sum_{i=1}^{N} \mathcal{L}_{\text{CE}}(f_{\theta+\hat{\theta}}(x_i), y_i) + \beta \cdot \max_{\tilde{x}_i \in \mathcal{A}_\varepsilon(x_i)} \text{KL}(f_{\theta+\hat{\theta}}(x_i) || f_{\theta+\hat{\theta}}(\tilde{x}_i))$$

(3)

$$\theta_\varepsilon = \underset{\theta}{\text{argmin}} \, \mathcal{L}_{\text{AWP},\theta,\varepsilon}$$

(4)

The model $f_{\theta_\varepsilon}$ which is the minimizer of the above loss, achieves an optimal trade-off between the cross-entropy loss on clean samples (clean loss, $\mathcal{L}_{\text{clean}}$) and weighted KL divergence between clean and adversarial images [25] (adversarial loss, $\mathcal{L}_{\text{adv}}$).

$$\mathcal{L}_{\text{clean},\theta} = \sum_{i=1}^{N} \mathcal{L}_{\text{CE}}(f_\theta(x_i), y_i), \quad \mathcal{L}_{\text{adv},\theta,\varepsilon} = \beta \cdot \sum_{i=1}^{N} \text{KL}(f_\theta(x_i) || f_\theta(\tilde{x}_i)), \quad \tilde{x} \in \mathcal{A}_\varepsilon(x)$$

(5)

Since the loss attains a minima at $\theta_\varepsilon$, a direction of further reduction in both clean and adversarial losses does not exist (Eq.4). Gradient descent on the adversarial loss at $\theta_\varepsilon$ to parameter $\theta'$ results in a reduction in adversarial loss at the cost of a higher clean loss.

$$\theta' = \theta_\varepsilon - \eta \cdot \nabla \mathcal{L}_{\mathrm{adv}, \theta_\varepsilon, \varepsilon} \tag{6}$$

$$\mathcal{L}_{\mathrm{adv}, \theta', \varepsilon} < \mathcal{L}_{\mathrm{adv}, \theta_\varepsilon, \varepsilon}, \quad \mathcal{L}_{\mathrm{clean}, \theta'} > \mathcal{L}_{\mathrm{clean}, \theta_\varepsilon} \tag{7}$$

We assume that $\mathcal{L}_{\mathrm{clean}, \theta_\varepsilon}$ increases by a rate $\gamma_{\mathrm{CE}, \theta_\varepsilon}$ and $\mathcal{L}_{\mathrm{adv}, \theta_\varepsilon, \varepsilon}$ decreases by a rate $\gamma_{\mathrm{KL}_\varepsilon, \theta_\varepsilon}$ in the direction of gradient descent of the adversarial loss ($-\nabla \mathcal{L}_{\mathrm{adv}, \theta_\varepsilon, \varepsilon}$). Since $\theta_\varepsilon$ is a minimizer of loss,

$$\mathcal{L}_{\mathrm{AWP}, \theta', \varepsilon} > \mathcal{L}_{\mathrm{AWP}, \theta_\varepsilon, \varepsilon} \implies \gamma_{\mathrm{CE}, \theta_\varepsilon} = \gamma_{\mathrm{KL}_\varepsilon, \theta_\varepsilon} + \alpha, \quad \alpha > 0 \tag{8}$$

Since the Cross-entropy loss is minimized at $x$, it attains a minima at $x$ within the pixel neighborhood.

$$\mathcal{L}_{\mathrm{CE}}(f_{\theta_\varepsilon}(x), y)) < \mathcal{L}_{\mathrm{CE}}(f_{\theta_\varepsilon}(x+\epsilon), y) \tag{9}$$

Since the model $f_\theta$ is adversarially robust within $\varepsilon$, the loss surface is locally smooth within the open interval $(-\varepsilon, \varepsilon)$. The adversarial loss, which measures the KL divergence of the adversarial image w.r.t. the clean image is least at $x$ and increases monotonically in the $\varepsilon$-neighborhood for adversarially robust models [3]. Therefore, adversarial loss at $\varepsilon' = \varepsilon + \delta$ for a small enough $\delta$ would be marginally higher than the loss at $\varepsilon$.

$$\mathcal{L}_{\mathrm{adv}, \theta_\varepsilon, \varepsilon'} > \mathcal{L}_{\mathrm{adv}, \theta_\varepsilon, \varepsilon} \tag{10}$$

Based on the local smoothness in weight space as well, the same property holds at any $\theta'$ in the gradient descent direction of $\mathcal{L}_{\mathrm{adv}, \theta_\varepsilon, \varepsilon}$ (satisfying Eq.6) such that $||\theta' - \theta_\varepsilon|| \leq \psi$, as shown below:

$$\mathcal{L}_{\mathrm{adv}, \theta_\varepsilon, \varepsilon'} = \mathcal{L}_{\mathrm{adv}, \theta_\varepsilon, \varepsilon} + \delta_{\theta_\varepsilon} \tag{11}$$

$$\mathcal{L}_{\mathrm{adv}, \theta', \varepsilon'} = \mathcal{L}_{\mathrm{adv}, \theta', \varepsilon} + \delta_{\theta'} \tag{12}$$

Since the adversarial loss at $\theta'$ is lower than the same at $\theta_\varepsilon$, the loss surface at $\theta'$ has a lower local Lipschitz constant when compared to $\theta_\varepsilon$ leading to the following:

$$\delta_{\theta'} < \delta_{\theta_\varepsilon} \implies \mathcal{L}_{\mathrm{adv}, \theta', \varepsilon'} - \mathcal{L}_{\mathrm{adv}, \theta', \varepsilon} < \mathcal{L}_{\mathrm{adv}, \theta_\varepsilon, \varepsilon'} - \mathcal{L}_{\mathrm{adv}, \theta_\varepsilon, \varepsilon} \tag{13}$$

Rearranging terms,

$$\mathcal{L}_{\mathrm{adv}, \theta_\varepsilon, \varepsilon'} - \mathcal{L}_{\mathrm{adv}, \theta', \varepsilon'} > \mathcal{L}_{\mathrm{adv}, \theta_\varepsilon, \varepsilon} - \mathcal{L}_{\mathrm{adv}, \theta', \varepsilon} \tag{14}$$

For small enough gradient descent step size $\eta$, the above can be related to the rate of reduction in the adversarial loss as shown below:

$$\gamma_{\mathrm{KL}_{\varepsilon'}, \theta_\varepsilon} = \gamma_{\mathrm{KL}_\varepsilon, \theta_\varepsilon} + \alpha', \quad \alpha' > 0 \tag{15}$$

Since the Cross-entropy loss does not depend on $\varepsilon$, from Eq.8 and Eq.15,

$$\gamma_{\mathrm{CE}, \theta_\varepsilon} - \gamma_{\mathrm{KL}_{\varepsilon'}, \theta_\varepsilon} = \alpha - \alpha' \tag{16}$$

Therefore, the rate of increase in $\mathcal{L}_{\mathrm{AWP}, \theta_\varepsilon, \varepsilon'}$, or $\alpha - \alpha'$, is less than the rate of increase in $\mathcal{L}_{\mathrm{AWP}, \theta_\varepsilon, \varepsilon}$, or $\alpha$. For small enough $\alpha$, $\alpha' > \alpha$, so that the overall loss $\mathcal{L}_{\mathrm{AWP}, \theta_\varepsilon, \varepsilon'}$ decreases. Based on this, for small enough $\eta$, $\exists\, \theta_{\varepsilon'}$ s.t. $||\theta_{\varepsilon'} - \theta_\varepsilon|| \leq \psi$ and $\mathcal{L}_{\mathrm{AWP}, \theta_{\varepsilon'}, \varepsilon'} < \mathcal{L}_{\mathrm{AWP}, \theta_\varepsilon, \varepsilon'}$. Hence, it is possible to move to $\theta_{\varepsilon'}$ which has a lower overall loss than the loss at $\theta_\varepsilon$. ∎

### E.4 Integrating ACAT with other efficient training methods

The proposed ACAT defense uses the KL divergence loss between clean and adversarial images, similar to the TRADES adversarial training algorithm [25]. We present results by integrating the proposed ACAT defense with losses from existing efficient adversarial training algorithms [19, 20] in Table-8. We obtain a significant boost in performance over the respective baselines, when we

**Algorithm 2** Ascending Constraint Adversarial Training (ACAT)

---

1: **Input:** Network $f_\theta$, Training Dataset $\mathcal{D} = \{(x_i, y_i)\}$, Adversarial Threat model: $\ell_\infty$ bound of radius $\varepsilon$, number of epochs E, Maximum Learning Rate $LR_{max}$, M training mini-batches of size $n$, Cross-entropy loss $\ell_{CE}$, Weight perturbation constraint $\mathcal{M}(\theta)$, coefficient of KL divergence term $\beta$
2: **for** $epoch = 1$ **to** $E$ **do**
3:     $\varepsilon_{asc} = epoch \cdot \varepsilon/E$
4:     $\mathrm{LR} = 0.5 \cdot \mathrm{LR}_{max} \cdot (1 + cosine((epoch - 1)/E \cdot \pi))$
5:     **for** $iter = 1$ **to** $M$ **do**
6:         **for** $i = 1$ **to** $n$ (in parallel) **do**
7:             **for** $steps = 1$ **to** 2 **do**
8:                 $\delta = 0.001 \cdot \mathcal{N}(0, 1)$
9:                 $\delta = \delta + \varepsilon_{asc} \cdot \text{sign}\left(\nabla_\delta \mathrm{KL}(f_\theta(x_i)||f_\theta(x_i + \delta))\right)$
10:                $\delta = Clamp\left(\delta, -\varepsilon_{asc}, \varepsilon_{asc}\right)$
11:                $\widetilde{x}_i = Clamp\left(x_i + \delta, 0, 1\right)$
12:            **end for**
13:        **end for**
14:        $\mathcal{L}_{\mathrm{TR}}(\theta) = \frac{1}{n}\sum\limits_{i=1}^{n}\mathcal{L}_{\mathrm{CE}}(f_\theta(x_i), y_i) + \beta \cdot \mathrm{KL}(f_\theta(x_i)||f_\theta(\tilde{x}_i))$
15:        $\tilde{\theta} = \underset{\hat{\theta} \in \mathcal{M}(\theta)}{\mathrm{argmax}}\, \mathcal{L}_{\mathrm{TR}}(\theta + \hat{\theta})$
16:        $\theta = \theta - \mathrm{LR} \cdot \nabla_\theta(\mathcal{L}_{\mathrm{TR}}(\theta + \tilde{\theta}))$
17:    **end for**
18: **end for**

---

use ACAT with GAT [19] and TRADES [25] losses, and a marginal boost when integrated with the NuAT defense [20]. The adversarial weight perturbation step in the proposed defense results in an increase in computational time when compared to the respective baselines. We choose the KL divergence based loss for both proposed defenses ACAT and DAJAT since it results in an optimal trade-off between performance and training time.

Table 8: **Integrating ACAT with different loss formulations** on the CIFAR-10 dataset with WideResNet-34-10 architecture. Robust accuracy is reported against the GAMA attack [19].

| Method | # Attack Steps | Clean Acc | Robust Acc | Time per epoch (seconds) |
|---|---|---|---|---|
| TRADES-AWP | 2 | 85.49 | 41.62 | 412 |
| ACAT (with TRADES loss) | 2 | 86.71 | 55.58 | 412 |
| NuAT2-WA | 2 | 86.32 | 55.08 | 334 |
| ACAT (with NuAT loss) | 2 | 86.19 | **55.91** | 530 |
| GAT2-WA | 2 | 87.36 | 50.24 | 267 |
| ACAT (with GAT loss) | 2 | **87.79** | 54.70 | 396 |

### E.5   Analysis of the effect of augmentations on ACAT

We analyze how using different combinations of augmentations affect ACAT in Table-9. From these results we can say that using hard augmentations like AutoAugment and a mix of base and AutoAugment each on 50% on the batch leads to a degradation in the performance of the model as compared to using simple augmentations like Pad+Crop+Horizontal Flip.

## F   Details on Experiments and Results

### F.1   Details on Datasets

We perform evaluations of the proposed defenses ACAT and DAJAT on the CIFAR-10, CIFAR-100 [13] and ImageNette-10 [12, 8] datasets, comprising of 10, 100 and 10 classes respectively. The resolution of images in the CIFAR-10 and CIFAR-100 datasets is 32x32, while it is 128x128 in

Table 9: Impact of augmentations: Performance (%) of ACAT models on Base augmentations and AutoAugment (Auto). Clean and robust accuracy against GAMA attack [19] are reported. The use of AutoAugment results in ∼ 1.5 - 2% drop in robust accuracy. The use of Base Augmentations alone (Pad+Crop+HFlip) gives the best overall performance on the unaugmented test set.

| Architecture | Train set | No-Aug (Clean) | No-Aug (Robust) | AutoAug (Clean) | AutoAug (Robust) |
|---|---|---|---|---|---|
| ResNet-18 | No-Augmentation | 73.50 | 43.64 | 44.98 | 18.50 |
|  | Base (Pad+Crop+HFlip) | 82.41 | 50.00 | 63.79 | 37.07 |
|  | AutoAugment | **82.54** | 48.11 | **76.40** | **43.22** |
|  | Base (50% batch) + AutoAugment (50% batch) | 81.15 | **50.01** | 70.89 | 40.93 |
| WideResNet-34-10 | No-Augmentation | 80.34 | 47.98 | 54.66 | 26.44 |
|  | Base (Pad+Crop+HFlip) | 86.71 | **55.58** | 68.24 | 40.83 |
|  | AutoAugment | **86.80** | 53.99 | **82.64** | **48.98** |
|  | Base (50% batch) + AutoAugment (50% batch) | 86.52 | 54.15 | 75.90 | 45.56 |

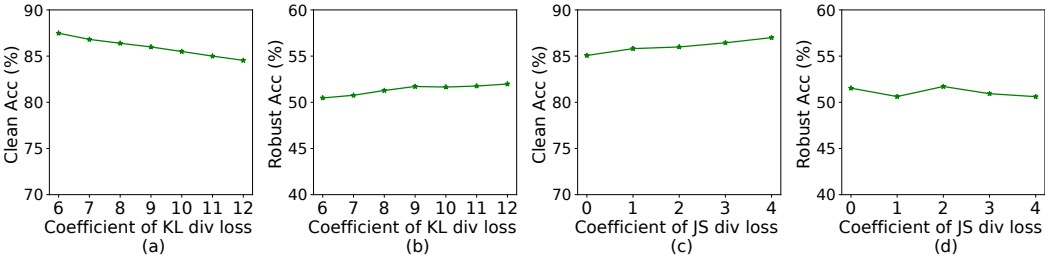

Figure 10: **Performance of DAJAT across variation in hyperparameters:** Performance of the proposed defense DAJAT (Base, 2*AA) on ResNet-18 architecture and CIFAR-10 dataset across variation in (a,b) coefficient of KL divergence loss between softmax outputs of clean and adversarial images. (c,d) coefficient of the JS divergence loss between the representations of different augmentations. Robust accuracy against GAMA [19] attack is shown.

ImageNette dataset. We use ResNet-18 [11] and WideResNet-34-10 [24] architectures for the CIFAR-10 and CIFAR-100 datasets and ResNet-18 architecture for ImageNette. While CIFAR-10 is the most popular dataset used for benchmarking adversarial defenses, CIFAR-100 has a larger number of classes with one-tenth the number of images in each class, making it a more challenging setting. ImageNette dataset is used to show the performance of the proposed method on higher resolution images. We consider an $\ell_\infty$ threat model with $\varepsilon = 8/255$ for the primary evaluations across all datasets. We use TRADES-AWP [23] [2] as the base code for most of our analysis. For analysis in Table-17 we use HAT [17] [3] and OAAT [1][4] as the base codes. For cutmix analysis in Table-15 we use [16] [5] as the base code. The license for each of these codes are available on their respective github repositories. Since the datasets used in this work are public, commonly used for research purposes and do not contain any objectionable content, we find no need to take any consent from the authors of these datasets.

### F.2 Details on Training Settings

The training for the baseline methods and the proposed approach is done for 110 epochs as is common in literature [15]. We use SGD optimizer along with cosine learning rate schedule for the proposed method with a momentum of 0.9. We fix the maximum learning rate to 0.2 and weight decay to 5e-4 for the proposed approach across all datasets and model architectures. For the baselines, we use the training settings from the official codes released by the authors. We performed all the experiments on two NVidia V100 GPUs, two RTX-2080 GPUs and one RTX-3090 GPU. We use a validation split of 1000 images from the train dataset both for CIFAR-10 and CIFAR-100 images, except for the results reported in Table-4 of the main paper, where we use the full dataset to compare against the state-of-the-art defenses on the RobustBench leaderboard [6].

---

[2] https://github.com/csdongxian/AWP

[3] https://github.com/imrahulr/hat

[4] https://github.com/val-iisc/OAAT

[5] https://github.com/imrahulr/adversarial_robustness_pytorch

Table 10: Performance gains obtained using DAJAT on CIFAR-100 using **larger capacity models**

| Training Algorithm | Architecture | Clean Accuracy | Robust Accuracy (GAMA) | Robust Accuracy (AutoAttack) |
|---|---|---|---|---|
| TRADES-AWP | | 62.73 | 29.92 | 29.59 |
| DAJAT (**Ours**) | WRN-34-10 | 68.74 | 31.58 | 31.30 |
| **Gains using DAJAT** | | **6.01** | **1.66** | **1.71** |
| TRADES-AWP | | 63.12 | 30.15 | 29.83 |
| DAJAT (**Ours**) | WRN-34-20 | 70.49 | 32.91 | 32.55 |
| **Gains using DAJAT** | | **7.37** | **2.76** | **2.72** |

Table 11: Comparison of DAJAT with Fixed-$\varepsilon$ and Ascending-$\varepsilon$ schedules on ResNet-18 and WideResNet-34-10 architectures on CIFAR-10 and CIFAR-100 datasets.

| Model architecture | Dataset | Epsilon schedule | Clean Acc | Robust Acc | Clean Acc (with Weight Averaging) | Robust Acc (with Weight Averaging) |
|---|---|---|---|---|---|---|
| **ResNet-18** | CIFAR-10 | Fixed $\varepsilon$ | 86.57 | 51.17 | 86.25 | 51.44 |
| | CIFAR-10 | Varying $\varepsilon$ (DAJAT) | 86.13 | 51.37 | 85.99 | 51.71 |
| | CIFAR-100 | Fixed $\varepsilon$ | 66.03 | 26.24 | 65.50 | 26.55 |
| | CIFAR-100 | Varying $\varepsilon$ (DAJAT) | 66.50 | 27.12 | 66.84 | 27.61 |
| **WRN-34-10** | CIFAR-10 | Fixed $\varepsilon$ | 91.46 | 31.01 | 90.09 | 47.30 |
| | CIFAR-10 | Varying $\varepsilon$ (DAJAT) | 89.12 | 56.98 | 88.90 | 57.22 |
| | CIFAR-100 | Fixed $\varepsilon$ | 71.04 | 19.90 | 70.60 | 25.97 |
| | CIFAR-100 | Varying $\varepsilon$ (DAJAT) | 68.82 | 30.75 | 68.74 | 31.58 |

For the proposed approaches ACAT and DAJAT (Base, 2*AA), we vary the value of $\beta$ to achieve an optimal trade-off between the clean and adversarial accuracy. As shown in Fig.9 and Fig.10(a,b), as we increase $\beta$, robust accuracy improves and clean accuracy degrades initially, with a saturating trend in robust accuracy at higher values of $\beta$. For the ResNet-18 model and CIFAR-10 dataset, the optimal value of $\beta$ is 8 and 9 for ACAT and DAJAT respectively.

We further fix the value of $\beta$ to the optimal setting of 9 and vary the coefficient of the JS divergence term in Fig.10(c,d). This term leads to a boost in the clean accuracy at the cost of a slight degradation in the robust accuracy. The optimal setting of the coefficient of JS divergence term is 2 in the given setting (Base, 2*AA) and ranges from 1 to 3 across all settings, datasets and model architectures.

We consider the Base, 2*AA as the main setting of DAJAT since its computational complexity matches with that of TRADES-AWP. However, we show the result of Base, 3*AA as well to highlight that the performance improves with a further increase in diversity. In this case, since the weight of the Base augmentations is considerably low in the overall loss (L16 in Algorithm-1), we give a weight of 1/3 to the TRADES loss on base augmentations and 2/3 to the TRADES loss on the AutoAugment based images. This mimics the setting of Base, 2*AA in terms of loss weighting, while introducing additional diversity due to the presence of a larger number of complex augmentations, yielding a small boost in performance.

### F.3 Performance on Larger Capacity Models

We find that the performance gains obtained using DAJAT are indeed higher on larger capacity models. In Table-10, we present results on CIFAR-100 dataset, on WideResNet-34-10 and WideResNet-34-20 model architectures using 110 epochs of training. We note that the improvements in clean accuracy increase from 6.01% to 7.37%, while the improvements in robust accuracy against AutoAttack increase from 1.71% to 2.72%.

### F.4 Impact of Ascending Perturbation Radius in DAJAT

We present an ablation of DAJAT without ascending perturbation radius (DAJAT+Fixed-$\varepsilon$), where attacks are constrained within a fixed perturbation bound of 8/255 during training. Table-11 shows results on CIFAR-10 and CIFAR-100 datasets with WideResNet-34-10 and ResNet-18 architectures. Robust Accuracy is reported against the GAMA attack. As seen in the first half of the table, on ResNet-18 architecture, there is no gradient masking even with DAJAT+Fixed-$\varepsilon$. However, on the WideResNet-34-10 architecture, DAJAT+Fixed-$\varepsilon$ results in a large drop in robust accuracy due to gradient masking. Although the use of weight averaging improves results, its robust accuracy is

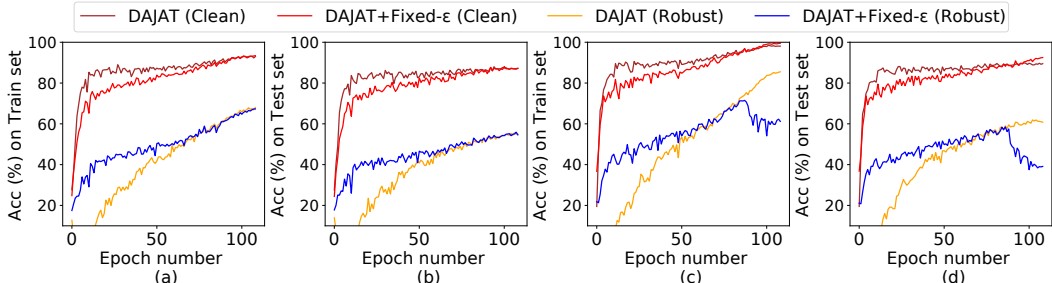

Figure 11: **Train and test Accuracy plots of DAJAT and DAJAT+Fixed-$\varepsilon$ to show impact of varying $\varepsilon$ training schedule:** The train and test plots of DAJAT are compared with an ablation experiment where a fixed $\varepsilon$ training schedule is used (DAJAT+Fixed-$\varepsilon$). Clean and Robust accuracy against PGD-20 [14] attack are plotted on CIFAR-10 dataset using ResNet-18 (a,b) and WideResNet-34-10 (c,d) architectures. Although DAJAT+Fixed-$\varepsilon$ is stable at lower model capacities (ResNet-18), there is a sudden drop in the robust accuracy due to Gradient masking on WideResNet-34-10 (c,d) after epoch 87, indicating the need for using varying $\varepsilon$ schedule in DAJAT. Evaluations are done using an attack perturbation bound of $\varepsilon = 8/255$.

Table 12: **Variance across reruns:** Variation in performance (%) of the proposed defenses ACAT and DAJAT on CIFAR-10 dataset and ResNet-18 architecture across three reruns. Based on the low standard deviation across runs, we note that both approaches are stable across reruns. Robust accuracy is evaluated against GAMA PGD-100 attack [19].

|  | ACAT | | DAJAT | |
| --- | --- | --- | --- | --- |
|  | Clean Acc | Robust Acc | Clean Acc | Robust Acc |
| Run-1 | 82.41 | 50.00 | 86.13 | 51.37 |
| Run-2 | 82.49 | 50.08 | 85.96 | 51.51 |
| Run-3 | 82.54 | 50.12 | 85.94 | 51.48 |
| Average | 82.48 | 50.07 | 86.01 | 51.45 |
| Standard Deviation | 0.07 | 0.06 | 0.1 | 0.07 |

still lower by around 5-10% when compared to DAJAT. The phenomenon of gradient masking in DAJAT+Fixed-$\varepsilon$ can also be observed in Fig.11, where its robust accuracy suddenly drops after epoch 87 accompanied by an increase in clean accuracy on CIFAR-10 with WideResNet-34-10 architecture. We note that even with respect to the best epoch accuracy on DAJAT+Fixed-$\varepsilon$, we obtain an improvement of 2.57% on robust accuracy using DAJAT.

### F.5   Variance across reruns

We present the variation across three reruns for the proposed defenses ACAT and DAJAT (Base, 2*AA) on CIFAR-10 dataset and ResNet-18 architecture in Table-12. Since weight averaging is known to improve the stability of the base method [20], we present results without the use of weight averaging to highlight the inherent variance of the base algorithm. The standard deviation of robust accuracy across reruns is low (0.06-0.07) across both approaches indicating stability of the proposed method. It can be noted that the standard deviation of clean accuracy in the proposed defense DAJAT is slightly higher (0.1) due to the randomness in the complex augmentations, which impacts clean accuracy more than the robust accuracy. Overall, we note that the standard deviation of clean and robust accuracy for both proposed defences ACAT and DAJAT is low .

### F.6   Analysis on Compute, Flops and Performance

We present the FLOPs, number of parameters and training time on a single Nvidia RTX-3090 GPU along with our results when compared to the TRADES-AWP baseline on the CIFAR-10 dataset for

Table 13: **Comparison of compute, FLOPs (per iteration) and performance** of the proposed approaches DAJAT and ACAT when compared to TRADES-AWP on CIFAR-10 dataset.

| Model | Method | gigaFLOPS (GFLOPS) | | Parameters (Million) | | Train Time | Clean Acc | Robust Acc |
|---|---|---|---|---|---|---|---|---|
| | | Training | Inference | Training | Inference | /epoch(sec) | (%) | (%) |
| ResNet18 | TRADES-AWP | 2707.6 | 71.251 | 11.174 | 11.174 | 299 | 80.47 | 49.87 |
| | ACAT (**Ours**) | 997.52 | 71.251 | 11.174 | 11.174 | 108 | 82.41 | 49.80 |
| | DAJAT (**Ours**) | 2422.5 | 71.251 | 11.184 | 11.174 | 264 | 85.99 | 51.48 |
| WRN-34-10 | TRADES-AWP | 32417 | 853.08 | 46.160 | 46.160 | 1633 | 85.10 | 55.87 |
| | ACAT (**Ours**) | 11943 | 853.08 | 46.160 | 46.160 | 472 | 86.71 | 55.36 |
| | DAJAT (**Ours**) | 29005 | 853.08 | 46.183 | 46.160 | 1381 | 88.90 | 56.96 |

Table 14: **Comparison of the proposed augmentation scheme with CutMix based augmentations [18]:** Performance (%) of the proposed defense DAJAT (Base, 2*AA) when compared to the use of CutMix based augmentation proposed by Rebuffi et al. [18] against PGD 40-step attack [14]

| Method | Clean Acc | Robust Acc (PGD-40) |
|---|---|---|
| TRADES [25] | 84.72 | 56.92 |
| Rebuffi et al. [18] | 87.24 | 57.60 |
| TRADES-AWP [23] | 85.35 | 59.13 |
| DAJAT (**Ours**) (Base, 2*AA) | **88.90** | **60.97** |

ResNet18 and WRN-34-10 models in Table-13. Robust accuracy is reported against AutoAttack. We discuss our observations below:

- FLOPs and number of parameters during inference are identical among the three training methods (TRADES-AWP, ACAT, DAJAT), since we use only a single batch-normalization layer (corresponding to Pad+Crop) during inference. As expected, these values are higher for WideResNet-34-10 model architecture when compared to ResNet-18.

- Since we use split batch-norm layers in DAJAT, the number of parameters increases by 0.05% during training, while it remains the same during inference.

- We compute FLOPs during training by considering that a single backward pass requires twice the number of FLOPs when compared to a forward pass. We also provide the number of forward and backward passes in each method for reference.

- Using ACAT, we achieve 63% reduction in FLOPs (training) and training time when compared to the TRADES-AWP baseline, while achieving 1.6-1.9% higher clean accuracy and comparable robust accuracy.

- The use of ACAT strategy in the proposed DAJAT defense enables us to achieve similar computational complexity as TRADES-AWP defense, while obtaining gains in performance. Using DAJAT, we achieve 10% reduction in FLOPs (training) and training time, while obtaining 3.8-5.5% higher clean accuracy and 1-1.6% higher robust accuracy.

## F.7 Comparison against CutMix based augmentation

While we compare the performance of the proposed approach against various base adversarial training algorithms [23, 14, 15, 20] in the main paper, we additionally compare with a recent augmentation scheme that uses CutMix augmentations [18] to improve performance in this section. The authors of [18] show a significant boost in performance using 400 epochs of training and large model architectures. However, to ensure a fair comparison, we report the result of 110 epochs of training on WideResNet-34-10 architecture and CIFAR-10 dataset that has been shared by the authors with us upon request. We report the PGD 40-step accuracy as shared by the authors. As shown in Table-14, we obtain a significant boost in performance over the CutMix based augmentation as well as the TRADES-AWP [23] baseline using the proposed defense DAJAT.

Additionally, contrary to the claims by Rebuffi et al. [18], we show that it is indeed possible to effectively use augmentations that modify the low-level statistics of images for obtaining improved performance in Adversarial Training by using the proposed defense DAJAT.

Table 15: Performance (%) by using [16] on CIFAR-10 dataset with Preact-ResNet18 model with Swish Activation trained using varying epsilon schedule and cosine learning rate unless specified otherwise.

| Method | Clean | GAMA |
|---|---|---|
| [C1]: Cutmix - step schedule + fixed eps | 81.67 | 49.18 |
| [C2]: Cutmix | **83.34** | 49.24 |
| [C3]: Ours(Base, Cutmix) | 82.67 | 51.99 |
| [C4]: Ours(Base, Cutmix, Cutmix) | 83.05 | **52.22** |
| [C5]: C1 with Relu | 81.03 | 46.6 |
| [C6]: C1 with Weight decay for BN | 70.66 | 36.36 |

Table 16: Performance of the proposed defense DAJAT when compared to some concurrent works on **CIFAR-10 and CIFAR-100 datasets** for ResNet18 and WideResNet-34-10 models. Robust evaluations are performed on Auto-Attack [5].

| Method | CIFAR-10, ResNet-18 | | CIFAR-10, WRN-34-10 | | CIFAR-100, ResNet-18 | | CIFAR-100, WRN-34-10 | |
|---|---|---|---|---|---|---|---|---|
| | Clean | AA @ 8/255 | Clean | AA @ 8/255 | Clean | AA @ 8/255 | Clean | AA @ 8/255 |
| AWP [23] | 81.99 | 51.45 | 85.36 | 56.17 | 59.88 | 25.81 | 62.73 | 29.59 |
| HAT [17] | 85.63 | 49.54 | 86.21 | 51.46 | 59.19 | 23.26 | 59.95 | 24.55 |
| SEAT [22] | 83.7 | 51.3 | 86.44 | 55.67 | 56.28 | **27.87** | - | - |
| SEAT+Cutmix [22] | 81.53 | 49.1 | 84.81 | 56.03 | - | - | - | - |
| TRADES + TE [9] | 83.86 | 49.77 | - | - | 59.35 | 25.27 | - | - |
| UDR + TRADES [4] | 84.4 | 49.9 | 84.93 | 54.45 | - | - | - | - |
| DAJAT (**Ours**) | **85.71** | **52.50** | **88.71** | **57.81** | **65.45** | 27.69 | **68.75** | **31.85** |

As present in the github repository [6] of [18] we note that naively using cutmix doesn't give good results as shown in Table-6 of the main paper, therefore as suggested we use [16] as the base code and incorporate cutmix into it. We present the results for 200 epochs training with learning rate drop of 0.1 at 100 and 150 epochs, using the PreActResNet-18 model with Swish Activation and batch size of 128 in Table-15(C1). We observe significantly improved results as compared to Table-6 of the main paper on using [16] as the base code. We observe that the key differences in [16] as compared to [25] are:

- Use of swish activation function in the PreActResNet18 model
- Weight decay not used for batch normalization layers

To study the impact of these changes, we investigate the use ReLU instead of Swish activation (Table-15(C5)) and the use of weight decay for all the parameters of the model including the batch normalization layers (Table-15(C6)). In both cases, we observe a significant drop with respect to C1. Thus based on this ablation, the use of swish activation, and avoiding weight decay for batch normalization layers seems important to obtain benefits using Cutmix.

Further we incorporate linearly increasing varying epsilon schedule along with cosine learning rate schedule and get improved results in Table-15(C2). Next we incorporated our method DAJAT with C2 and present the results in Table-15(C3,C5), where we can observe significant gains over C2, thus showing the effectiveness of DAJAT.

### F.8 Comparison of DAJAT with Concurrent Works

Here we compare the proposed approach DAJAT trained for 200 epochs with recent works that appeared at ICLR 2022. The comparison of the proposed method DAJAT with AWP [23], HAT [17], Self Ensemble Adversarial Training (SEAT) [22] with (SEAT+cutmix) and without cutmix (SEAT), Unified distributional robustness for TRADES (UDR + TRADES) [4], temporal ensembling with TRADES (TRADES + TE) [9] on CIFAR10 and CIFAR100 datasets for ResNet18 and WideResNet-34-10 models is shown in Table-16. Since these are very recent works, we only present results reported in the paper, and leave the remaining entries in the table blank.

---

[6]https://github.com/deepmind/deepmind-research/tree/master/adversarial_robustness/pytorch

Table 17: Performance (%) of DAJAT when **combined with other Adversarial training methods**, OAAT [1] and HAT [17] on **CIFAR-10 and CIFAR-100** with 110 epochs of training. Robust evaluations are performed on Auto-Attack(AA) [5] at $\varepsilon = 8/255$ and $16/255$.

| Method | CIFAR-10, ResNet-18 | | | CIFAR-10, WRN-34-10 | | | CIFAR-100, ResNet-18 | | | CIFAR-100, WRN-34-10 | | |
|---|---|---|---|---|---|---|---|---|---|---|---|---|
| | Clean | AA, 8/255 | AA, 16/255 | Clean | AA, 8/255 | AA, 16/255 | Clean | AA, 8/255 | AA, 16/255 | Clean | AA, 8/255 | AA, 16/255 |
| AWP [25, 23] | 80.47 | 49.87 | **19.23** | 85.10 | 55.87 | **23.27** | 59.88 | 25.81 | 8.28 | 62.73 | 29.59 | **11.04** |
| AWP+DAJAT | 85.99 | 51.48 | 16.33 | 88.90 | 56.96 | 19.73 | 66.84 | 27.32 | 8.97 | 68.74 | 31.30 | 9.91 |
| OAAT [1] | 80.24 | 50.88 | 22.05 | 85.67 | 55.93 | 24.05 | 61.70 | 26.77 | 9.91 | 65.73 | 30.35 | 12.01 |
| OAAT+DAJAT | 82.05 | 52.21 | 22.78 | 86.22 | 57.64 | 24.56 | 62.50 | 28.47 | 10.67 | 66.03 | 31.15 | 12.67 |
| HAT [17] | 85.63 | 49.54 | 14.96 | 86.21 | 51.46 | **16.76** | 59.19 | 23.26 | 6.96 | 59.95 | 24.55 | 7.13 |
| HAT+DAJAT | 86.68 | 51.47 | 16.38 | 86.71 | 53.85 | 16.50 | 62.78 | 26.49 | 8.72 | 64.88 | 27.37 | 8.71 |

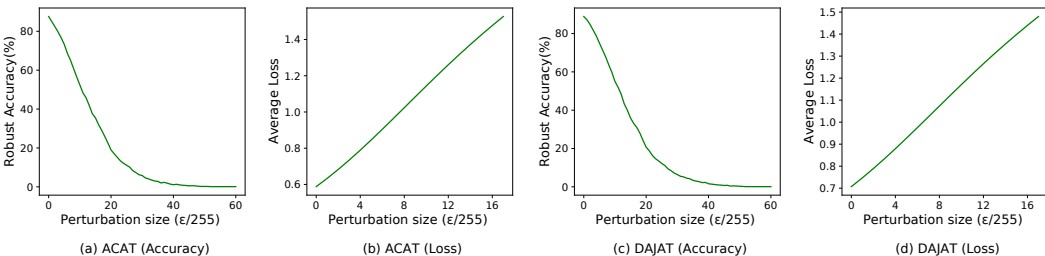

(a) ACAT (Accuracy)  (b) ACAT (Loss)  (c) DAJAT (Accuracy)  (d) DAJAT (Loss)

Figure 12: **Robust Accuracy and Loss on FGSM attack against variation in perturbation size:** (a,c) Robust accuracy (%) of the proposed defenses ACAT and DAJAT against PGD 7-step attacks across variation in attack perturbation bound. Attacks within larger perturbation bounds are able to bring down the robust accuracy of the model to 0, indicating the absence of gradient masking. (b,d) Cross-entropy loss on FGSM adversarial samples across variation in attack perturbation bound. The linearly increasing trend of loss indicates the absence of gradient masking. The models are trained on CIFAR-10 dataset using ResNet-18 architecture.

## F.9  Combining the proposed approach with different adversarial training methods

We explore combining the proposed defense DAJAT with some existing methods like [23], [1] and [17] in Table-17. We observe that combining DAJAT with all three existing works leads to significant gains both in clean as well as adversarial accuracies (AA, 8/255), especially on CIFAR-100 where the number of images per class is low. Although OAAT [1] shows improved results over AWP [23], combining DAJAT with OAAT leads to further gains of $\sim 1.5\%$ in both clean and adversarial accuracy on CIFAR10 and $1 - 1.5\%$ gains in both clean and adversarial accuracy on CIFAR100. Further, since OAAT [1] claims to achieve robustness at larger epsilon bounds, we evaluate using Auto-Attack at $\varepsilon = 16/255$. Using OAAT+DAJAT we observe gains over OAAT on AutoAttack with $\varepsilon = 16/255$ as well, which further confirms the effectiveness of DAJAT. Finally we combine DAJAT with HAT [17] and we observe consistent gains over HAT [17] on all models and datasets. While HAT [17] proposes to improve the robustness-accuracy trade-off, combining DAJAT with HAT further improves this trade-off and shows gains of $\sim 1\%$ on clean accuracy and $\sim 2\%$ on robust accuracy for CIFAR-10, and $3 - 5\%$ on clean accuracy and $\sim 3\%$ on robust accuracy for CIFAR-100 dataset.

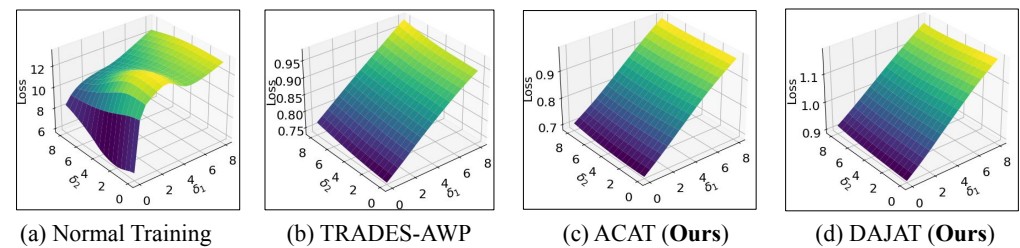

(a) Normal Training  (b) TRADES-AWP  (c) ACAT (**Ours**)  (d) DAJAT (**Ours**)

Figure 13: **Loss Surface Plots:** Plot of cross-entropy loss in the local neighborhood of images along the gradient direction ($\delta_1$) and a random direction perpendicular to the gradient ($\delta_2$). The loss surface of the proposed defenses ACAT and DAJAT are smooth similar to the TRADES-AWP defense, indicating the absence of gradient masking.

Table 18: **Evaluation against Black-Box and White-Box FGSM [10] attacks and multi-step PGD attacks [14]:** Performance (%) of the proposed DAJAT (Base + 2*AA) defense on CIFAR-10 dataset with ResNet-18 architecture

| Method | Clean Acc | BB FGSM | FGSM | PGD-20 | PGD-100 | PGD-500 |
|---|---|---|---|---|---|---|
| NuAT2-WA | 86.32 | 84.71 | 63.48 | 58.09 | 57.74 | 57.74 |
| ACAT | 86.71 | 85.29 | 64.08 | 58.76 | 58.64 | 58.53 |
| TRADES-AWP | 85.36 | 83.93 | 63.49 | 59.22 | 59.11 | 59.08 |
| DAJAT(Base, 3*AA ) | **88.64** | **87.19** | **66.99** | **61.09** | **60.80** | **60.74** |

Table 19: **Evaluation against multi-step Targeted and Untargeted PGD attacks [14] with single and multiple random restarts:** Performance (%) of the proposed defense DAJAT (Base, 2*AA) across different datasets with ResNet-18 architecture

| | CIFAR-10 | | CIFAR-100 | | IN-10 | |
|---|---|---|---|---|---|---|
| **Attack** | 500-step | 1000-step | 500-step | 1000-step | 500-step | 1000-step |
| PGD-Targeted (Least Likely Class) | 85.01 | 85.01 | 66.02 | 65.98 | 85.06 | 85.01 |
| PGD-Targeted (Random Class) | 80.56 | 80.55 | 63.96 | 63.96 | 80.13 | 80.13 |
| PGD-Untargeted | 55.21 | 55.20 | 32.89 | 32.89 | 65.07 | 65.07 |
| | 1-RR | 1000-RR | 1-RR | 1000-RR | 1-RR | 1000-RR |
| PGD 50-step, r-RR | 55.30 | 54.55 | 32.98 | 32.09 | 65.20 | 65.02 |

### F.10    Sanity checks to verify the absence of gradient masking

We perform several sanity checks as recommended by Athalye et al. [3] to ensure the absence of gradient masking in the proposed defenses ACAT and DAJAT.

- From Table-18 we note that Black-Box attacks are weaker than White-Box attacks, indicating that the gradients from the model are reliable.

- We further note from Table-18 that attacks with higher number of steps are stronger than those with lower steps. Further, PGD multi-step attacks are stronger than FGSM white-box attacks.

- From Table-19 we note that robust accuracy against targeted and untargeted attacks saturates as the number of attack steps increase from 500 to 1000, indicating that the evaluation is robust.

- We also note from Table-19 that the drop in accuracy with 1000 random restarts is marginal.

- We note from Fig.12 that an increase in perturbation bound increases the effectiveness of PGD 7-step attacks, and is able to bring down the accuracy of the model to 0 at large bounds. Further, the loss on FGSM samples monotonically increases in the vicinity of the data samples. These trends indicate the absence of gradient masking.

- We present results against AutoAttack [5] in Tables-2 and 3 of the main paper. AutoAttack is an ensemble of several gradient-based attacks and a gradient-free attack Square [2]. The robust accuracy against AutoAttack is similar to the accuracy against gradient-based attack GAMA [19] indicating that gradient-free attacks are not significantly stronger than gradient based attacks.

- We show the loss surface plots of the proposed defenses ACAT and DAJAT in the vicinity of data samples in Fig.13. We note that the loss surface of the proposed defenses is smooth similar to the TRADES-AWP defense, indicating the absence of gradient masking.

We finally compare the robust accuracy against various attacks in Tables-18 and 19 with the robust accuracy against GAMA attack [19] and AutoAttack [5] in Tables-2 and 3 of the main paper. The latter evaluations are significantly stronger, indicating that the evaluation presented in the main paper is robust.