# OpenReview forum: "Efficient and Effective Augmentation Strategy for Adversarial Training"
_NeurIPS.cc/2022/Conference — NeurIPS 2022 Accept_

### Official Review · Reviewer_v5Go · 2022-07-10

**Rating:** 7
**Confidence:** 5
**Soundness:** 4 excellent
**Presentation:** 3 good
**Contribution:** 4 excellent

**Summary:**

The paper proposes an augmentation strategy for adversarial training by combining simple and complex augmentations to improve the effectiveness of the augmentation and introducing ACAT that gradually increases adversarial epsilon with 2 adversarial defence steps to allow the method to work efficiently. Three claims of justification are listed based on the connection with domain generalization theory. Two important conjectures are proposed and verified through detailed analysis and solid experiments. Comprehensive experiments conducted on several benchmarks for comparisons and ablation studies are provided in the main paper and the supplementary materials. With comprehensive experiments and substantial improvement, this study may be an important milestone for future work.

**Questions:**

NA

**Strengths And Weaknesses:**

Strengths:

-The paper is well-written and easy to follow. Claims are supported by detailed analysis and convincing experiments.

-The proposed method gains substantial improvement over the state-of-the-art baselines and beats the concurrent work.

-The proposed method is novel, effective, efficient and convincing. The paper provides better understanding of augmentations in adversarial training.

-The authors conduct comprehensive experiments considering different settings and different comparisons to verify their ideas.

Weakness:

-The author claims that the method is efficient, could they provide FLOPs, number of parameters and training time of the model?

---

> ### Author Response · Authors · 2022-08-02
> **Response to Reviewer v5Go**
>
> We sincerely thank the reviewers for their time and valuable feedback on our work. We are happy to see that the reviewers find our work clear and well-written [*nzVH*, *7Xuh*, *v5Go*], efficient [*Kbj7*, *v5Go*] and effective [*Kbj7*, *nzVH*, *v5Go*], well-supported by thorough experimentation [*nzVH*, *v5Go*] and analysis [*Kbj7*, *nzVH*, *v5Go*], and the results noteworthy [*Kbj7*, *nzVH*, *7Xuh*, *v5Go*]. It is indeed very encouraging that *nzVH* and *v5Go* mention that our work would be very useful to the community, and could be an important milestone and source of inspiration for future work. We will incorporate the reviewers' suggestions in our camera-ready version.
>
> **Addressing comments specific to reviewer *v5Go***
>
> We sincerely thank the reviewer for the encouraging feedback on our work.
>
> **FLOPs, number of parameters and training time**
>
> We present the FLOPs, number of parameters, training time on a single Nvidia 3090 GPU and results of our proposed approaches ACAT and DAJAT when compared to the TRADES-AWP baseline on the CIFAR-10 dataset in the following tables. Robust accuracy is reported against AutoAttack.
>
> We present results on the WideResNet-34-10 model architecture in the below table:
>
> |   Method   | FLOPs (inference) |       Number of Forward passes       |       Number of Backward passes      | FLOPs (training) | Number of Parameters (training) | Number of Parameters (inference) | Training Time/epoch (sec) | Clean Acc (%) | Robust Acc (%) |
> |:----------:|:-----------------:|:------------------------------------:|:------------------------------------:|:----------------:|:---------------------------------:|:----------------------------------:|:-------------------------:|:-------------:|:--------------:|
> | TRADES-AWP |     8.5308E+11    | 10(Attacks) + 2(AWP) + 2(Train) = 14 | 10(Attacks) + 1(AWP) + 1(Train) = 12 |    3.2417E+13    |             4.6160E+07            |             4.6160E+07             |            1633           |      85.10     |      55.87     |
> |    ACAT (Ours)   |     8.5308E+11    |  2(Attacks) + 2(AWP) + 2(Train) = 6  |  2(Attacks) + 1(AWP) + 1(Train) = 4  |    1.1943E+13    |             4.6160E+07            |             4.6160E+07             |            472            |     86.71     |      55.36     |
> |    DAJAT (Ours)    |     8.5308E+11    |  6(Attacks) + 2(AWP) + 6(Train) = 14 |  6(Attacks) + 1(AWP) + 3(Train) = 10 |    2.9005E+13    |             4.6183E+07            |             4.6160E+07             |            1381           |      88.90     |      56.96     |
>
>
> We present results on the ResNet-18 model architecture in the below table:
>
>
> |   Method   | FLOPs (inference) |       Number of Forward passes       |       Number of Backward passes      | FLOPs (training) | Number of Parameters (training) | Number of Parameters (inference) | Training Time/epoch (sec) | Clean Acc (%) | Robust Acc (%) |
> |:----------:|:-----------------:|:------------------------------------:|:------------------------------------:|:----------------:|:---------------------------------:|:----------------------------------:|:-------------------------:|:-------------:|:--------------:|
> | TRADES-AWP |     7.1251E+10    | 10(Attacks) + 2(AWP) + 2(Train) = 14 | 10(Attacks) + 1(AWP) + 1(Train) = 12 |    2.7076E+12    |             1.1174E+07            |             1.1174E+07             |            299            |     80.47     |      49.87     |
> |    ACAT (Ours)   |     7.1251E+10    |  2(Attacks) + 2(AWP) + 2(Train) = 6  |  2(Attacks) + 1(AWP) + 1(Train) = 4  |    9.9752E+11    |             1.1174E+07            |             1.1174E+07             |            108            |     82.41     |      49.80      |
> |  DAJAT (Ours)    |     7.1251E+10    |  6(Attacks) + 2(AWP) + 6(Train) = 14 |  6(Attacks) + 1(AWP) + 3(Train) = 10 |    2.4225E+12    |             1.1184E+07            |             1.1174E+07             |            264            |     85.99     |      51.48     |

---

> > ### Author Response · Authors · 2022-08-02
> > **Response to Reviewer v5Go (2)**
> >
> > We discuss our observations below:
> > - FLOPs and number of parameters during inference are identical among the three training methods (TRADES-AWP, ACAT, DAJAT), since we use only a single batch-normalization layer (corresponding to Pad+Crop) during inference. As expected, these values are higher for WideResNet-34-10 model architecture when compared to ResNet-18.
> > - Since we use split batch-norm layers in DAJAT, the number of parameters increases by 0.05% during training, while it remains the same during inference.
> > - We compute FLOPs during training by considering that a single backward pass requires twice the number of FLOPs when compared to a forward pass. We also provide the number of forward and backward passes in each method for reference.
> > - Using ACAT, we achieve 63% reduction in FLOPs (training) and training time when compared to the TRADES-AWP baseline, while achieving 1.6-1.9% higher clean accuracy and comparable robust accuracy.
> > - The use of ACAT strategy in the proposed DAJAT defense enables us to achieve similar computational complexity as TRADES-AWP defense, while obtaining gains in performance. Using DAJAT, we achieve 10% reduction in FLOPs (training) and training time, while obtaining 3.8-5.5% higher clean accuracy and 1-1.6% higher robust accuracy.
> >
> > We believe this discussion on FLOPs, number of parameters and training time would be a valuable addition to our submission. We will include this in the final version.
> >
> > We will be happy to answer any further questions related to our work.

---

> > > ### Comment · Reviewer_v5Go · 2022-08-09
> > > **Response to the authors**
> > >
> > > Thank you for your response. The rebuttal is strong. I would like to keep my score unchanged.

---

> > > > ### Author Response · Authors · 2022-08-10
> > > > **Response to Reviewer v5Go**
> > > >
> > > > We thank the reviewer for the reply and the valuable feedback.

---

### Official Review · Reviewer_7Xuh · 2022-07-11

**Rating:** 5
**Confidence:** 3
**Soundness:** 3 good
**Presentation:** 3 good
**Contribution:** 2 fair

**Summary:**

This paper combines many previous robust training techniques together: adversarial training, Auto Augmentation, Split batch normalization, smoothness regularization (JSD loss), progressively hardened adversarial training, etc.

**Questions:**

Please see above.

**Limitations:**

Please see above.

**Strengths And Weaknesses:**

Strengths:
The paper is well written, and the method achieves good empirical performance.

Weakness:
1. The technical novelty is limited.
This proposed method is just a simple combination of many previous robust training techniques together: adversarial training, Auto Augmentation, Split batch normalization, smoothness regularization (JSD loss in AugMix), progressively hardened adversarial training [1,2], etc. It is undoubtedly a good technical report but hard to justify its technical novelty as a research paper.

[1] Curriculum Adversarial Training. IJCAI, 2018.
[2] Improving Adversarial Robustness Through Progressive Hardening. 2020.

2. Unfortunately, I find some claims in the paper to be wrong.

For example, in lines 211-214: "Prior works use separate batch normalization layers for clean and adversarial images to improve the performance of adversarial training, specifically in a self-supervised learning scenario [46, 27, 14]. However, in a supervised learning setting, state-of-the-art defenses often use single batch normalization layers for both clean and adversarial images."

This is wrong. Unfortunately, your paper is not the first to use separate BN in supervised learning. In fact, separate BN is originally proposed and most widely used in supervised learning. Just to name a few: [3,4,5,6,7,8]

[3] Intriguing properties of adversarial training. ICLR, 2020.
[4] Adversarial examples improve image recognition. CVPR, 2020.
[5] Does data augmentation benefit from split batchnorms. 2020.
[6] Once-for-all adversarial training: In-situ tradeoff between robustness and accuracy for free. NeurIPS, 2020.
[7] Shape-texture debiased neural network training. ICLR, 2021.
[8] AugMax: Adversarial composition of random augmentations for robust training. NeurIPS, 2021.

---

> ### Author Response · Authors · 2022-08-02
> **Response to Reviewer 7Xuh (2)**
>
> 2) **Incorrect claims in the paper**
>
> We would like to clarify what we mean in the following statement:
>
> "*Prior works use separate batch normalization layers for clean and adversarial images to improve the performance of adversarial training, specifically in a self-supervised learning scenario [46, 27,213 14]. However, in a supervised learning setting, state-of-the-art defenses often use single batch normalization layers for both clean and adversarial images.*"
>
> - "*Prior works use separate batch normalization layers for clean and adversarial images to improve the performance of adversarial training*"
>     -  Here we intend to acknowledge that while we do use split batch-norm layers in our algorithm, we **do not** claim to be the first to use it.
>
> - "*specifically in a self-supervised learning scenario*"
>    - Here we intend to say that specifically in the setting of self-supervised Adversarial Training, split batch-norm is seen to be very important, resulting in large performance gains.
>
> - "*However, in a supervised learning setting, state-of-the-art defenses often use single batch normalization layers for both clean and adversarial images.*"
>    - We would like to clarify that here we **do not** claim that we are the first to use split batch-norm in a supervised learning setting.
>    - We emphasize on "*state-of-the-art defenses*" and "*often*" - We intend to say that **most** defenses which are **currently state-of-the-art** (for example top ones on the RobustBench leaderboard) do not use split batch-norm layers. We mention this to show that split batch-norm in a supervised **Adversarial Training** setting is not indispensable.
>
> We understand that the above statement was possibly confusing, which led the reviewer to misinterpret that we claimed to be the first to use split batch-norm in a supervised learning setting. We have updated this in the revised submission.
>
> We thank the reviewer for all the citations provided, which we have incorporated in our revised version (Lines 211-212, 234 in the revised submission).
>
> **We request the reviewer to kindly re-evaluate our work in light of the above clarification.**
>
> We will be happy to answer any further questions related to our work.

---

> ### Author Response · Authors · 2022-08-02
> **Response to Reviewer 7Xuh**
>
> We sincerely thank the reviewers for their time and valuable feedback on our work. We are happy to see that the reviewers find our work clear and well-written [*nzVH*, *7Xuh*, *v5Go*], efficient [*Kbj7*, *v5Go*] and effective [*Kbj7*, *nzVH*, *v5Go*], well-supported by thorough experimentation [*nzVH*, *v5Go*] and analysis [*Kbj7*, *nzVH*, *v5Go*], and the results noteworthy [*Kbj7*, *nzVH*, *7Xuh*, *v5Go*]. It is indeed very encouraging that *nzVH* and *v5Go* mention that our work would be very useful to the community, and could be an important milestone and source of inspiration for future work. We will incorporate the reviewers' suggestions in our camera-ready version.
>
> **Addressing comments specific to reviewer *7Xuh***
>
>  1) **Novelty**
>
> We believe the notion of novelty is very subjective, and respect the reviewer's opinion on this with respect to our work. However, for the purpose of assessment at NeurIPS, we would like to justify our stance in light of the "Originality" definition provided in the NeurIPS 2022 reviewer guidelines as quoted below:
>
> "***Originality: Are the tasks or methods new? Is the work a novel combination of well-known techniques? (This can be valuable!) Is it clear how this work differs from previous contributions? Is related work adequately cited***"
>
> We believe our work is indeed "***a novel combination of well-known techniques***". We list below the key contributions of our work, to justify "***how this work differs from the previous contributions***":
>
>  - While it is well-known that data augmentations improve the performance of standard training, naively using augmentations in adversarial training is known to degrade performance. To understand this, we firstly present a conjecture to clearly state the role of augmentations in training Deep Networks, and using this we analyse how adversarial training differs from standard training. We support the same using empirical results and citations from prior work as well. [Sections 4.1, 4.2]
>  - We propose DAJAT, that is a novel combination of several well-known methods as pointed by the reviewer - adversarial training, AutoAugment, split batch-norm, JS divergence, increasing $\varepsilon$ schedule. While the use of split batch normalization is not new for OOD robustness and adversarial training, we are the first to use split batch-norm **for different augmentations** (and not for clean/adversarial images) in the context of **adversarial training**. We refer the reviewer to Fig. 14 in the revised submission, which clarifies the use of split batch-norm in the proposed method w.r.t. existing adversarial training methods.
>
>  - While an increasing $\varepsilon$ schedule has been used in adversarial training to obtain robustness at larger perturbation bounds [39, 1], we are the first to use this to build an efficient two-step defense ACAT. Existing efficient adversarial training methods deploy complex techniques such as additional smoothing regularizers, or adaptive strategies to detect the onset of gradient masking. We show analytically and empirically that a simple combination of increasing $\varepsilon$ schedule, weight-space smoothing and cosine learning rate schedule can indeed lead to performance gains at lower compute (Section 5.4). This is in fact a generic strategy and we show that it applies to several other methods as well (Section A.3).
>
> - Our method has a wide applicability and can combined with many adversarial training strategies (as shown in Table-13) and multiple augmentations (as shown in Section-6.3 and Table-6).
>
> - Lastly, we would like to highlight our empirical contributions, which clearly distinguishes us from prior work (Table-4) and concurrent work (Table-12) as well. We obtain large gains over existing methods on several datasets and achieve remarkable gains in a low data scenario (CIFAR-100, Imagenette) where data augmentations are most effective. On CIFAR-100, we outperform all existing methods on the RobustBench leaderboard, including the ones that utilize additional training data.
>
>
>
>
> "***Is related work adequately cited***"
>
> - Our work combines various aspects of existing works, and we have ensured that we cite existing works for every aspect that we use - adversarial training [31,51,45], AutoAugment [9], Split batch normalization [46,27,14], smoothness regularization [21], varying $\varepsilon$ schedule [39,1]. Kindly note that the citations correspond to the original submission and not the revision, to justify that we indeed cite related work sufficiently.
>
> - As rightly summarized by the reviewer *nzVH*,
> "*This work starts from previous results in the literature to come up with a simple but effective approach regarding the use of data augmentation for adversarial training*"

---

> > ### Comment · Reviewer_7Xuh · 2022-08-03
> > **Follow-up response**
> >
> > Thank you for posting this careful response. Unfortunately, I'm still not convinced this paper is novel.
> >
> > For sure I agree with the NeurIPS 2022 reviewer guidelines that "a novel combination of well-known techniques can be valuable". There are plenty such good works, which simply combines previous methods and successfully solving an important problem.
> >
> > However, the key is whether you find the "hidden gem" or not.
> >
> > For example, if there are two well-known works on two different fields. People have been using them separately in their own fields, and no one ever realized it can bring much more benefit when combined together. One day someone suddenly realized this and publish a paper on it. Then this kind of simple combination is valuable. Because it uncovers the "hidden gem" which have been forever laying there waiting for people's attention.
> >
> > However, in this paper, all techniques combined share the same goal: to improve model robustness. And the paper combines them for the same purpose. This is different from the previous case. In this case, the combination is not a "hidden gem". It is too obvious and straightforward. To get better practical results, maybe the most straightforward thing to think about is to stack a bunch of techniques sharing the same purpose together, just like done in this paper.
> >
> > Given that, I don't think this paper is novel and will keep my original evaluation.
> >
> > With that said, I do think this is a good technical report and have practical meaning for machine learning engineering. But academic conference proceeding is not the best destination for it.

---

> > > ### Author Response · Authors · 2022-08-03
> > > **Response to reviewer 7Xuh's reply**
> > >
> > >
> > > We thank the reviewer *7Xuh* for the quick reply. This is indeed an interesting discussion, which not only pertains to our work, but to a larger issue that the ML community is facing today - Insightful papers seem to be getting lost in the trap of novelty, while many "novel" papers do not seem to find any utility other than getting into the long list of related works. **We hope to make this an open discussion and invite other reviewers and ACs to also share their views.**
> > >
> > > We completely agree with the reviewer on the view of "finding the hidden gem". We hope to convince the reviewer that we have indeed found a hidden gem.
> > >
> > > We request the reviewer to kindly consider the following scenario -
> > >
> > > Let us say, the community believes that a very well-known method ***xyz*** cannot be used for solving the extremely important problem ***abc*** unless some restricted conditions ***def*** are met. Now there is a submission at NeurIPS which challenges this belief, presents an analysis on why it is challenging to incorporate ***xyz*** to solve ***abc***, proposes to use a simple combination of existing works to enable ***xyz*** to be a useful and practical option to solve ***abc***, and gives sound (experimental) evidence that the method works well currently, and is also a "generic strategy" that can be integrated with future works.
> > >
> > > Would this be worthy of acceptance at NeurIPS?
> > >
> > > Here is what the ***xyz***'s and ***abc***'s correspond to in our work -
> > >
> > > - ***xyz*** : data augmentations
> > > - ***abc*** : Achieving adversarial robustness within $\ell_p$ norm bounds
> > > - ***def*** : Models need to have large capacity, augmentations need to preserve low-level features, training method is computationally expensive
> > >
> > >
> > > While it is well-known that data augmentations improve the performance of standard training, naively using augmentations in adversarial training is known to degrade performance. To understand this, we firstly present a conjecture to clearly state the role of augmentations in training Deep Networks, and using this we analyse how adversarial training differs from standard training in the context of data augmentations. We propose DAJAT that uses a combination of simple and complex augmentations with separate batch-norm layers to overcome the limitations associated with augmentations in adversarial training. We further propose to overcome the additional computational overhead associated with the use of multiple augmentations by using 2 attack steps for training. While existing works propose to use different training losses or adaptive training strategies to mitigate gradient masking, we show that a simple combination of well-known techniques, a varying epsilon schedule, weight space smoothing and cosine learning rate schedule, can effectively mitigate gradient masking. We also present an analysis on why this method (ACAT) is expected to work effectively. Overall, the proposed approach DAJAT combined with the benefits of ACAT is seen to work effectively across several benchmark datasets, outperforming other defenses on the RobustBench leaderboard. We show that DAJAT not only works with the TRADES-AWP loss formulation, but can be integrated with several other defenses and augmentations to obtain performance gains.
> > >
> > > It is unfortunate that the reviewer thinks of our work as an "obvious and straightforward" way of "stacking" a "bunch of techniques" that were anyway designed to solve the same goal of "improving robustness". We respectfully disagree with this, and present our view against what the reviewer mentions, using some excerpts from the other reviews as well:
> > >
> > > - [*7Xuh*] "***However, in this paper, all techniques combined share the same goal: to improve model robustness. And the paper combines them for the same purpose.***"
> > >     -  We would like to clarify that our goal is not merely to "improve model robustness" but to understand why data augmentations do not work in improving adversarial robustness, and to find a simple, efficient and effective way to accomplish the same. This has indeed been understood and appreciated by other reviewers as shown below:
> > >          - "*This work starts from previous results in the literature to come up with a simple but effective approach regarding the use of data augmentation for adversarial training.*"
> > >          - "*previous papers have shown that data augmentations are helpful for adversarial robustness for certain conditions: network with big capacity and data augmentations have to preserve low-level features. Here, this paper shows that these conditions can become unnecessary by using data augmentation specific batch normalization layers.*"
> > >          - "*The paper proposes an augmentation strategy for adversarial training by combining simple and complex augmentations to improve the effectiveness of the augmentation and introducing ACAT that gradually increases adversarial epsilon with 2 adversarial defence steps to allow the method to work efficiently.*"

---

> > > > ### Author Response · Authors · 2022-08-03
> > > > **Response to reviewer 7Xuh's reply (2)**
> > > >
> > > > - [*7Xuh*] "***It is too obvious and straightforward. To get better practical results, maybe the most straightforward thing to think about is to stack a bunch of techniques sharing the same purpose together, just like done in this paper.***"
> > > >    - If this was the case, it is hard to understand why the community still struggled to incorporate data augmentations in adversarial training. The insights that our paper brings have been appreciated by other reviewers as shown below:
> > > >       - "*The paper provides better understanding of augmentations in adversarial training.*"
> > > >       -  "*This type of paper with many ablation studies and in-depth analysis is very useful for the community as it can be a great source of inspiration for future work.*"
> > > >       - "*Experimental results speak for themselves with clear improvement on robust and clean accuracies.*"
> > > >       - "*With comprehensive experiments and substantial improvement, this study may be an important milestone for future work.*"
> > > >       - "*Claims are supported by detailed analysis and convincing experiments.*"
> > > >       - "*The proposed method is novel, effective, efficient and convincing.*"
> > > >
> > > > On a side note, we hope our response to the second issue (Incorrect claims in the paper) has been sufficiently convincing. We will be happy to discuss more and clarify any further concerns.
> > > >
> > > > We wish to pose this question to ponder on - to solve a given problem, is it better to find a simple and effective solution, or to find a complex and "novel" solution that may not even generalize well to all use cases?

---

> > > ### Comment · Reviewer_7Xuh · 2022-08-07
> > > **Response to the authors**
> > >
> > > Thank you for your response.
> > >
> > > [5] shows that separate batch normalization between weak and strong augmentations (which has very similar definitions as in your paper: they use AA as the strong augmentation) can help improve accuracy on clean samples. Adversarial training is widely known to benefit adversarial robustness. So, a naive way to improve both clean accuracy and adversarial robustness is to combine [5] with adversarial training (e.g., TRADES), as done in the proposed method.
> > >
> > > To me this is super intuitive and straightforward. But I agree this might be subjective, although I believe I'm not intentionally doing so. Others in this field may find this inspiring. And given the good empirical performance, I think this paper is beneficial to the community. After all, who would say no to a new state-of-the-art method on the robustness benchmark.
> > >
> > > Given that, I increase my rating but lower my confidence score since my updated evaluation might be subjective.

---

> > > > ### Author Response · Authors · 2022-08-09
> > > > **Response to Reviewer 7Xuh**
> > > >
> > > > We thank the reviewer for reconsidering the contributions of our work and updating the score.
> > > >
> > > > We will update the final version to highlight the various other contributions of our work (listed in our earlier replies) in addition to the training algorithm. We will also include a more detailed comparison with [5] in the final version, although it is not published as on date.
> > > >
> > > > We would like to add a quick comment that many observations in our work are contrary to the findings in [5] suggesting that the underlying phenomenon is different, as explained in the respective papers. For example, from the ablations table in [5], we note that the use of single batch-norm (instead of split batch-norm) degrades the accuracy by only 0.2% suggesting that this is not absolutely essential in the standard ERM setting, as we also note in Section-4.

---

### Official Review · Reviewer_nzVH · 2022-07-11

**Rating:** 6
**Confidence:** 4
**Soundness:** 4 excellent
**Presentation:** 4 excellent
**Contribution:** 3 good

**Summary:**

This paper tackles the problem of Lp-norm robustness for classification. More specifically, previous papers have shown that data augmentations are helpful for adversarial robustness for certain conditions: network with big capacity and data augmentations have to preserve low-level features. Here, this paper shows that these conditions can become unnecessary by using data augmentation specific batch normalization layers. In this way, complex data augmentations will not impact the network capacity. The authors also propose an optimised training strategy with: 1) increasing perturbation radius throughout training and 2) using JS divergence.

**Questions:**

* Please see above the weaknesses section: especially point 1) which could be easily addressed.

* Just a quick related work comment: split batch normalization layers have been used for supervised learning setting (page 3) in AdvProp and similar line of work (Pyramid Adversarial Training) and not only self-supervised learning setting.

**Limitations:**

The authors did not mention any negative societal impact of their work.

**Strengths And Weaknesses:**

Strengths:
* The paper is very clear and well-written. This work starts from previous results in the literature to come up with a simple but effective approach regarding the use of data augmentation for adversarial training.
* Very thorough experimental section and appendix. This type of paper with many ablation studies and in-depth analysis is very useful for the community as it can be a great source of inspiration for future work.
* Experimental results speak for themselves with clear improvement on robust and clean accuracies.

Weaknesses:
1) The usefulness of the ascending perturbation radius could be more extensively checked in the paper. Currently, in Figure 2, panel (a) shows better performance on clean accuracy but not really on robust accuracy in panel (b) as the robust accuracy at 8/255 seems the same when training at 8/255 with or without ACAT.  Hence, it is not clear that it is a key element in DAJAT as there is no ablation study of DAJAT without ascending perturbation radius (sorry if I missed it) and it would have been nice to have in Table 2 ACAT vs fixed perturbation radius (besides just having Figure 2 for this comparison).
2) It is not clear whether this approach stills helps for larger architectures as split batch normalization layers is already a way of cleverly increasing the network expressiveness with few parameters. Maybe, this approach has diminishing returns when the network has more capacity. This would require further investigation but I understand that it might require too much compute to check it.

---

> ### Author Response · Authors · 2022-08-02
> **Response to Reviewer nzVH**
>
> We sincerely thank the reviewers for their time and valuable feedback on our work. We are happy to see that the reviewers find our work clear and well-written [*nzVH*, *7Xuh*, *v5Go*], efficient [*Kbj7*, *v5Go*] and effective [*Kbj7*, *nzVH*, *v5Go*], well-supported by thorough experimentation [*nzVH*, *v5Go*] and analysis [*Kbj7*, *nzVH*, *v5Go*], and the results noteworthy [*Kbj7*, *nzVH*, *7Xuh*, *v5Go*]. It is indeed very encouraging that *nzVH* and *v5Go* mention that our work would be very useful to the community, and could be an important milestone and source of inspiration for future work. We will incorporate the reviewers' suggestions in our camera-ready version.
>
> **Addressing comments specific to reviewer *nzVH***
>
> We thank the reviewer for the encouraging comments and an accurate summary of our work.
>
> 1) **Usefulness of the ascending perturbation radius at 8/255**
>
> Ascending perturbation radius helps in mitigating gradient masking when lesser steps are used for attack generation. While the impact of an ascending radius is lesser in low capacity models such as ResNet-18, its importance is more evident in large capacity models such as WideResNet-34-10. We present results of  ACAT when compared to Fixed constraint AT (TRADES-AWP with 2 attack steps and $\varepsilon=8/255$) on CIFAR-10 dataset with ResNet-18 and WideResNet-34-10 model architectures in the following table:
>
> | Architecture     | Method                  | Clean@last epoch | Robust@last epoch | Clean@best epoch | Robust@best epoch | Clean (Last - Best) | Robust Acc (Last - Best) |
> |------------------|-------------------------|:----------------:|:-----------------:|:----------------:|:-----------------:|:-------------------:|:------------------------:|
> | ResNet-18        | Fixed constraint AT     |       80.63      |       49.63       |       80.82      |       49.61       |        -0.19        |           0.02           |
> | ResNet-18        | Ascending constraint AT |       82.41      |       50.00       |       82.57      |       49.91       |        -0.16        |           0.09           |
> | WideResNet-34-10 | Fixed constraint AT     |       86.69      |       44.87       |       86.83      |       54.76       |        -0.14        |         **-9.89**        |
> | WideResNet-34-10 | Ascending constraint AT |       86.71      |       55.58       |       86.30      |       55.46       |         0.41        |           0.12           |
>
> Robust Accuracy is reported against the GAMA attack. Best accuracy is computed using PGD-20 attack, which is not very reliable. Hence, in some cases, best epoch may have a slightly lower accuracy when compared to the last epoch. On ResNet-18, we observe that the difference between last and best epochs for both methods is very low. However, on WideResNet-34-10, we observe the phenomenon of gradient masking in Fixed Constraint AT, with robust accuracy dropping by around 10% towards the end of training. We note that the difference between last and best epochs is very low for ACAT even on WideResNet-34-10. The main motivation of using an ascending perturbation radius is to stabilize training and prevent the onset of gradient masking.

---

> > ### Author Response · Authors · 2022-08-02
> > **Response to Reviewer nzVH (2)**
> >
> > 2) **Study of DAJAT without ascending perturbation radius**
> >
> > We present an ablation of DAJAT without ascending perturbation radius (DAJAT+Fixed-$\varepsilon$) where attacks are constrained within a fixed perturbation bound of 8/255 during training. The following table shows results on CIFAR-10 and CIFAR-100 datasets with WideResNet-34-10 and ResNet-18 architectures.
> >
> > | Model architecture | Dataset   | $\varepsilon$  schedule             | Clean Acc | Robust Acc | Clean Acc (with Weight Averaging) | Robust Acc (with Weight Averaging) |
> > |--------------------|-----------|-------------------------------|:---------:|:----------:|:---------------------------------:|:----------------------------------:|
> > | ResNet-18          | CIFAR-10  | DAJAT+Fixed-$\varepsilon$          |   86.57   |    51.17   |               86.25               |                51.44               |
> > | ResNet-18          | CIFAR-10  | Varying $\varepsilon$ (DAJAT) |   86.13   |    51.37   |               85.99               |                51.71               |
> > | ResNet-18          | CIFAR-100 | DAJAT+Fixed-$\varepsilon$          |   66.03   |    26.24   |               65.50               |                26.55               |
> > | ResNet-18          | CIFAR-100 | Varying $\varepsilon$ (DAJAT) |   66.50   |    27.12   |               66.84               |                27.61               |
> > | WideResNet-34-10          | CIFAR-10  | DAJAT+Fixed-$\varepsilon$          |   91.46   |    31.01   |               90.09               |                47.30               |
> > | WideResNet-34-10          | CIFAR-10  | Varying $\varepsilon$ (DAJAT) |   89.12   |    56.98   |               88.90               |                57.22               |
> > | WideResNet-34-10          | CIFAR-100 | DAJAT+Fixed-$\varepsilon$          |   71.04   |    19.90   |               70.60               |                25.97               |
> > | WideResNet-34-10          | CIFAR-100 | Varying $\varepsilon$ (DAJAT) |   68.82   |    30.75   |               68.74               |                31.58               |
> >
> > Robust Accuracy is reported against the GAMA attack. As seen in the first half of the table, on ResNet-18 architecture, there is no gradient masking even with DAJAT+Fixed-$\varepsilon$. However, on WideResNet-34-10 architecture, DAJAT+Fixed-$\varepsilon$ results in a large drop in robust accuracy due to gradient masking. Although the use of weight averaging improves results, its robust accuracy is still lower by around 5-10% when compared to DAJAT. The phenomenon of gradient masking in DAJAT+Fixed-$\varepsilon$ can also be observed in Fig.16 of the revised submission, where its robust accuracy suddenly drops after epoch 87 accompanied by an increase in clean accuracy on CIFAR-10 with WideResNet-34-10 architecture. We note that even with respect to the best epoch accuracy on DAJAT+Fixed-$\varepsilon$, we obtain an improvement of 2.57% robust accuracy using DAJAT.
> >
> > We acknowledge that the plot in Fig.2(a,b) indeed does not give a clear understanding of why ascending $\varepsilon$ training is important at 8/255. We will include the above results in the final version for better clarity.

---

> > > ### Author Response · Authors · 2022-08-02
> > > **Response to Reviewer nzVH (3)**
> > >
> > > 3) **Do the performance gains obtained using DAJAT reduce as model capacity increases?**
> > >
> > > The gains obtained using DAJAT are a result of allowing the network to learn different function mappings for different augmentations, while sharing majority of the parameters between the two. Due to the use of split batch normalization layers, model capacity increases merely by 0.05% during training (as shown in our reply to reviewer *v5Go*). There is no change in model capacity during inference. Therefore, we do not expect diminishing returns as model capacity increases. In fact, models with higher capacity benefit more from data augmentations, and hence we expect to obtain larger gains using DAJAT.
> > >
> > > We find that the performance gains obtained using DAJAT are indeed higher on larger capacity models. In the following table, we present results on CIFAR-100 dataset, on WideResNet-34-10 and WideResNet-34-20 model architectures using 110 epochs of training:
> > >
> > > | Training algorithm    | Model Architecture | Clean Accuracy | Robust Accuracy (GAMA) | Robust Accuracy (AutoAttack) |
> > > |-----------------------|--------------------|:--------------:|:----------------------:|:----------------------------:|
> > > | TRADES-AWP            | WRN-34-10          |      62.73     |          29.92         |             29.59            |
> > > | DAJAT (Ours)          | WRN-34-10          |      68.74     |          31.58         |             31.30            |
> > > | **Gains using DAJAT** | **WRN-34-10**      |    **6.01**    |        **1.66**        |           **1.71**           |
> > > | TRADES-AWP            | WRN-34-20          |      63.12     |          30.15         |             29.83            |
> > > | DAJAT (Ours)          | WRN-34-20          |      70.49     |          32.91         |             32.55            |
> > > | **Gains using DAJAT** | **WRN-34-20**      |    **7.37**    |        **2.76**        |           **2.72**           |
> > >
> > > We note that the improvements in clean accuracy increase from 6.01% to 7.37%, while the improvements in robust accuracy against AutoAttack increase from 1.71% to 2.72%.
> > >
> > > 4) We thank the reviewer for the references, which we will include in our submission.
> > >
> > > We will be happy to answer any further questions related to our work.

---

> ### Author Response · Authors · 2022-08-09
> **A gentle reminder to Reviewer nzVH**
>
> We would like to post a gentle reminder to the reviewer nzVH to kindly let us know if our rebuttal addresses the concerns posted. We will be happy to answer any further questions as well.

---

### Official Review · Reviewer_Kbj7 · 2022-07-18

**Rating:** 6
**Confidence:** 2
**Soundness:** 2 fair
**Presentation:** 3 good
**Contribution:** 3 good

**Summary:**

The authors group common data augmentation into two groups: simple and complex.
The authors propose to use separate batch normalization layers for the two types of data augmentations, simple and complex. They argue that this helps to reconcile the distribution shift between the two types of augmentations. They also propose minimizing the JSD between the output of a network with perturbed weights when given different augmentation of a sample as an input. They show improvements in the clean and robust accuracy on multiple datasets.

**Questions:**

Please address the points listed under Weaknesses and Questions above.

**Strengths And Weaknesses:**

Strengths:

- The authors' approach shows emperically gains in robustness and accuracy on multiple datasets.
- The methods work well on small datasets.
- The method is relatively computationally efficient.
- The authors try to explain the results using the transfer learning theory proposed in [4].



Weaknesses and Questions:
1. The paper is not very easy to follow.
2. The distintion between simple and complex is not very clear. For example: flip results in a large jump in the pixel space. Pad and crop is similar to translate in AA. I'm not sure how the authors draw the line and sperate the two augmentations. Many AA have little effect on the pixel space statistics.
3. Line 143-151:  I don't think the results shown conclusively support the arguments presented. I think the section should be revised to better explain what the authors mean. For example, the authors say (Line 145) `There is a large difference between the augmented
 data and test data in pixel space, although they may be similar in feature space. `. I think the differnce between the augmented data and the  test data is similar to that between the augmented data and the adverserial test data in the pixel space, since the adversarial examples are only an $\epsilon$ away from the real test data.
4. Table 1 should include baseline models trained with neither of the data augmentation types and both.
5. Lines 221-223: I don't understand the reasoning here. $\gamma$ and $\beta$ are learned parameters and expected to be different when trained on different data, even if the data disterbution is similar. $\gamma$ and $\beta$  scale and add a bias to the normalized data (equation in line 208). Figure 1 shows the the mean and var are the same for both augmentation types (in cos similarity) which hints the the statstics of both augmentation types are similar. This contracdicts the argument the authors presenting ?

**UPDATE:**
I appreciate the authors' comprehensive response and clarification of the ambiguous points. I must admit that I am still not convinced that the authors' method of splitting the augmentation is completely sound. However, the authors provide strong empirical evidence to support their approach.

---

> ### Author Response · Authors · 2022-08-02
> **Response to Reviewer Kbj7**
>
> We sincerely thank the reviewers for their time and valuable feedback on our work. We are happy to see that the reviewers find our work clear and well-written [*nzVH*, *7Xuh*, *v5Go*], efficient [*Kbj7*, *v5Go*] and effective [*Kbj7*, *nzVH*, *v5Go*], well-supported by thorough experimentation [*nzVH*, *v5Go*] and analysis [*Kbj7*, *nzVH*, *v5Go*], and the results noteworthy [*Kbj7*, *nzVH*, *7Xuh*, *v5Go*]. It is indeed very encouraging that *nzVH* and *v5Go* mention that our work would be very useful to the community, and could be an important milestone and source of inspiration for future work. We will incorporate the reviewers' suggestions in our camera-ready version.
>
> **Addressing comments specific to reviewer *Kbj7***
>
>  1) We will certainly work towards improving the clarity of our paper. We clarify the points highlighted by the reviewer below.
>
>  2) **Distinction between simple and complex augmentations**
>
>    We term the augmentations that preserve low-level features of images as simple augmentations, and those that modify the same as complex augmentations. To distinguish between simple and complex augmentations, we **do not** use the difference between two images in pixel-space, since this would incorrectly show that simple changes like horizontal-flip and crop are far apart, as the reviewer highlighted. Instead, we use metrics that better capture low-level features at pixel and patch levels.  This can be measured at a pixel-level using MSE between color histograms, and at a patch-level using patch-wise MSE. To compute patch-wise MSE between two images $x_1$ and $x_2$, for every $8\times8$ patch in $x_1$ we find the nearest patch in $x_2$ and a horizontal flip of $x_2$, and compute an average across all patches in $x_1$. We report the mean and standard deviation of this value across all images in the test set. We show the pair-wise distances (as mean ± standard deviation) between three sets of images (Unaugmented, Pad+Crop+HFlip, AutoAugment) in the following table -
>
> |             Image pairs             | MSE between Histograms  | MSE between Patches |
> |:-----------------------------------:|:----------------------:|:-------------------:|
> | Base (Pad+Crop+HFlip) / Unaugmented |     133.60 ± 94.05    |    43.68 ± 23.37   |
> |      AutoAugment /  Unaugmented     |    289.25 ± 405.11    |    51.39 ± 24.23   |
>
>
>    In the above table, lower value indicates that the images are more similar. We note that Pad+Crop+HFlip augmentations have the advantage of being more similar to the distribution of unaugmented images that are expected during inference, while AutoAugment transformed images are farther away from the unaugmented images. As rightly pointed out by the reviewer, AutoAugment consists of several augmentations of varying complexity levels, and may (rarely) contain augmentations of similar complexity as pad+crop as well. This is reflected in the higher variance in pair-wise distances corresponding to AutoAugment as shown in the above table.
>
>    DAJAT allows separate function mappings for augmentations that resemble the inference-time distribution (Pad+Crop+HFlip), and those that lead to better diversity (AutoAugment). Base augmentations have low variation, and are similar to the distribution of unaugmented images, which is important to obtain performance gains using the batch-norm layer corresponding to these base augmentations during inference. On the other hand, the high variance of AutoAugment based transformations helps in improving the robust generalization of the overall model.
>
>
>
> 3) **[L143-151] Improve clarity on the distribution shift between adversarial images**
>
>   In lines 143-151, we aim to compare the impact of augmentations on natural (clean) and adversarial images. We compare the shift between (natural-augmented, natural-unaugmented) pairs and (adversarial-augmented, adversarial-unaugmented) pairs. We consider two types of distances between image pairs: low-level (MSE between histograms/patches) and feature-level (FID). As rightly pointed out by the reviewer, in terms of low-level distances, we expect the distances between (natural-augmented, natural-unaugmented) pairs and the corresponding (adversarial-augmented, adversarial-unaugmented) pairs to be similar, since the perturbed images are only an $\varepsilon$ away from natural images. However, as seen in Fig.8,9,10, the perturbations of Pad+Crop+HFlip look similar to the perturbations of unaugmented images, while the perturbations of AutoAugment based images look different from those of unaugmented images. This is a result of larger pixel-level differences between the (natural-AutoAugment, natural-unaugmented) image pairs when compared to (natural-PadCrop, natural-unaugmented) image pairs, which serves as a more diverse initialization for the attack.

---

> > ### Author Response · Authors · 2022-08-02
> > **Response to Reviewer Kbj7 (2)**
> >
> > The difference in the absolute perturbations results in a larger distance in feature space (Fréchet Inception Distance or FID) as shown in the following table:
> >
> >   |             Image pairs             | FID between natural image pairs | FID between Adversarial image pairs |
> > |-----------------------------------|:-------------------------------:|:-----------------------------------:|
> > | Base (Pad+Crop+HFlip) / Unaugmented |              24.02              |                33.41                |
> > |      AutoAugment /  Unaugmented     |              37.62              |                43.75                |
> >
> >
> >
> >   For better clarity, we summarize our findings from the above mentioned tables and figures below:
> >
> >
> > | Natural/ Adversarial |        Distributions        | Low-level distance | Feature-level distance |
> > |--------------------|---------------------------|:------------------:|:----------------------:|
> > |    Natural images    |   Pad+Crop vs. Unaugmented  |   Low    |     Low     |
> > |    Natural images                  | Autoaugment vs. Unaugmented |   High   |     Medium      |
> > |   Adversarial images  |   Pad+Crop vs. Unaugmented  |   Low    |     Medium     |
> > |     Adversarial images                 | Autoaugment vs. Unaugmented |   High  |     **High**    |
> >
> > Our discussion in Line 143-151 of the paper relates to the bolded entry in the above table, that is, the higher feature level distance between (adversarial-AutoAugment, adversarial-unaugmented) image pairs when compared to (natural-AutoAugment, natural-unaugmented) image pairs. This relates to higher $\frac{1}{2}d_{\mathcal{F}\Delta\mathcal{F}}(s,t)$ in Eq.1 of the paper.
> >
> > 4) **Including more baseline results in Table-1**
> >
> > We present additional results on ACAT by training without any augmentations, and by training using a combination of both augmentations in every minibatch in the table below:
> >
> > |   Architecture   |                 Train set                  | No-Aug (Clean Acc) | No-Aug (Robust Acc) | AutoAug (Clean Acc) | AutoAug (Robust Acc) |
> > |----------------|------------------------------------------|:--------------:|:---------------:|:---------------:|:----------------:|
> > |     ResNet-18    |               No-Augmentation              |      73.50     |      43.64      |      44.98      |       18.50      |
> > |     ResNet-18    |            Base (Pad+Crop+HFlip)           |      82.41     |      50.00      |      63.79      |       37.07      |
> > |     ResNet-18    |                 AutoAugment                |      82.54     |      48.11      |      76.40      |       43.22      |
> > |     ResNet-18    | Base (50% batch) + AutoAugment (50% batch) |      81.15     |      50.01      |      70.89      |       40.93      |
> > | WideResNet-34-10 |               No-Augmentation              |      80.34     |      47.98      |      54.66      |       26.44      |
> > | WideResNet-34-10 |            Base (Pad+Crop+HFlip)           |      86.71     |      55.58      |      68.24      |       40.83      |
> > | WideResNet-34-10 |                 AutoAugment                |      86.80     |      53.99      |      82.64      |       48.98      |
> > | WideResNet-34-10 | Base (50% batch) + AutoAugment (50% batch) |      86.52     |      54.15      |      75.90      |       45.56      |
> >
> > Robust accuracies are reported against GAMA attack. We note that the use of Base Augmentations alone (Pad+Crop+HFlip) still gives the best overall performance on the unaugmented test set. We will update this in the final version.
> >
> > 5) **[L221-223] Clarification on Batch-Normalization running statistics and affine parameters**
> >
> > **(Comment by reviewer) Figure 1 shows that the mean and var are the same for both augmentation types (in cos similarity) which hints that the statistics of both augmentation types are similar**
> >
> > We find that although the cosine similarities between the running mean and variance of the two batch-norm layers in DAJAT are close to 1, they are not exactly equal to 1, as shown in Fig.15 of the paper revision. This small difference in running statistics indeed results in the performance gains that we observe. To validate this, we perform three ablation experiments - first by training with separate running statistics and common affine parameters, second by training with common running statistics and separate affine parameters, and third without using split batch-norm at all.

---

> > > ### Author Response · Authors · 2022-08-02
> > > **Response to Reviewer Kbj7 (3)**
> > >
> > > |                                  Method                                  | Clean Accuracy | Robust Accuracy |
> > > |------------------------------------------------------------------------|:--------------:|:---------------:|
> > > |       [E1] split running statistics + split affine parameters (Ours)        |      88.90      |      57.22      |
> > > |        [E2]    split running statistics + common affine parameters           |      88.61     |      56.91      |
> > > |        [E3]    common running statistics + split affine parameters           |      88.86     |      57.01      |
> > > | [E4] common running statistics + common affine parameters (single Batch-Norm) |      89.08     |      53.86      |
> > >
> > > Robust Accuracy is reported against the GAMA attack. As shown in the above table, E2 and E3 (having either split running statistics or split affine parameters) perform similar to the proposed approach, where separate running statistics and affine parameters are used. This reiterates the fact that our method learns a different function mapping for both augmentations, and this can be realized by having different running statistics or different affine parameters or using a combination of both. We further note that the use of a single batch-norm layer for both augmentations (E4) degrades results significantly.
> > >
> > > We compare the average cosine similarity of the running statistics across training iterations for both our method, and the case where we have separate running statistics and common affine parameters in Fig.15 of the paper revision. We note that the scale and trend of the average cosine similarity is similar in both cases. In the case where common affine parameters are used, the gains in results w.r.t. single batch-norm case can be attributed to the small drop in cosine similarity over training. This indicates that small changes in these running statistics can indeed lead to a large impact in the overall results (as shown in the above table). We further verify this by noting the large difference in performance of the models when different batch-norm parameters are used for training, in the table below. Robust Accuracy is reported against the GAMA attack.
> > >
> > > | Method                                                    | Batch-norm layer used for inference | Clean Acc   | Robust Acc  |
> > > |-----------------------------------------------------------|:-----------------------------------:|:-----------:|:-----------:|
> > > | split running statistics + split affine parameters (Ours) |            Pad+Crop+HFlip           |     88.9    |    57.22    |
> > > | split running statistics + split affine parameters (Ours) |             AutoAugment             |    76.17    |    44.45    |
> > > | split running statistics + common affine parameters       |            Pad+Crop+HFlip           |    88.61    |    56.91    |
> > > | split running statistics + common affine parameters       |             AutoAugment             |    78.69    |    45.41    |
> > >
> > >
> > >
> > > We thank the reviewer for their insightful comments, which will certainly add value to our paper. We will include the above discussion in our final version. We will be happy to answer any further questions related to our work.

---

> ### Author Response · Authors · 2022-08-09
> **Response to the update by Reviewer Kbj7**
>
> We sincerely thank the reviewer for the reply and increase in score.
>
> We provide further justification on the method of splitting augmentations, since we believe that a sound understanding of why this works is also crucial in addition to strong empirical evidence.
>
>
> In the proposed method, we use two sets of augmentations - Pad+Crop+HFlip and AutoAugment. As mentioned in L97 and L614, the second set of augmentations (complex augmentations) consists of an autoaugment based transformation followed by the base augmentations (Pad+Crop+HFlip). This has two implications - firstly, this ensures that the complexity of these augmentations is always greater than or equal to the base augmentations. Secondly, since AutoAugment returns the unaugmented image as well, with a certain probability (0.22 for CIFAR-10 policy), the base augmentations form a subset of the complex augmentations. This trend is indeed reflected in the distribution of pair-wise feature-level similarities (Cosine similarity between features obtained from an Inception-V3 network) between the following pairs: (Unaugmented, Pad+Crop+HFlip (PCHf)) and (Unaugmented, AutoAugment+PCHf) plotted in Fig.17(a) of the revised submission. When AutoAugment alone is applied, a large fraction of images have a very high cosine similarity, while others have a more spread out distribution, as shown in Fig.17(b). When Pad+Crop+HFlip is applied in series, the distribution of these images shifts to the left, leading to an overlap in the two distributions.
>
> Now, the following question arises: with such a large overlap in distributions, why does the method work?
>
> The role of the "complex" batch-norm layer is to allow the learning of a function that minimizes empirical risk across a wide distribution of data. While the test distribution may be different from these augmentations, learning from diverse data is known to prevent overfitting and improve generalization. However, since the task of adversarial training is inherently hard, and the objective of minimizing loss on a wider distribution of data makes the task harder, we observe a drop in overall accuracy. The use of a separate batch-norm layer for "simple" augmentations allows the network to **specialize** on a select subset that is close to the distribution of test set images, and has a low variance. Although the diversity of simple augmentations is low, it is sufficient to learn the batch-norm statistics and affine parameters which constitute 0.05% of all parameters, while the majority of the parameters are learned using both distributions, resulting in low overfitting.
>
>
> We hope this clarifies the concerns of the reviewer. We will be happy to clarify any further concerns as well. Since the author response window closes today, we will include clarifications on any further concerns directly in the final version.

---

### Meta-Review · Area_Chair_zt9b · 2022-08-27

**Recommendation:** Accept
**Confidence:** Certain

**Metareview:**

The reviewers found the paper well written and were satisfied with the experimental setting, which shows clear improvements. The authors made a thorough rebuttal and carefully answered the reviewer's questions and I recommend for acceptance as I believe this will be useful to the community.

I recommend the authors to carefully go over the reviewers’ comments and incorporate them into the final manuscript, along with the additional experiments from the rebuttal.


**Award:**

No

---

### Decision · Program_Chairs · 2022-09-14

Accept